# Neural Diversity Regularizes Hallucinations in Language Models

**Kushal Chakrabarti**                                   *kushalc@obviouslywrong.org*
**Nirmal Balachundhar**                                  *nbalachundhar@gmail.com*
*South Park Commons, San Francisco, CA*

**Reviewed on OpenReview:** *https://openreview.net/forum?id=5l9ZflyApA*

## Abstract

Language models continue to hallucinate despite scaling parameters, compute, and data. We propose *neural diversity* — decorrelated parallel representations — as a provable mechanism to reduce hallucination rates at fixed parameter and data budgets. While existing mitigation strategies largely target accuracy, we *reframe it as a second-moment reliability problem* governed by representational covariance and *provide the first formal tail bounds* for hallucination probability in ensembled language models, explaining 94.3% of reliability variation across configurations in our setting (Qwen2.5-0.5B, 20M Pile tokens, 12 tasks). We introduce ND-LoRA (Neural Diversity Low-Rank Adaptation), combining parallel LoRA adapters with Barlow Twins regularization, and *reduce hallucinations by up to 25.6% (and 14.6% on average)* while preserving capability. Ablations show LoRA and regularization act synergistically; causal interventions identify neural diversity as the mediating factor; correlational studies indicate scale, with +0.1% neural correlation associated with +3.8% hallucination. Finally, task-dependent optima emerge: different tasks require different optimal diversity. Neural diversity enables reliability gains without scaling, improving tails orthogonally to parameters and data at near-zero cost (+0.008% pretraining, 1.1× latency).

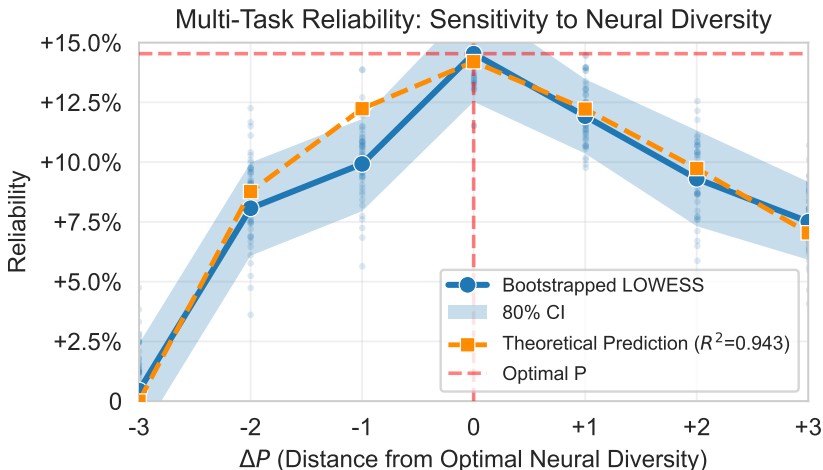

Figure 1: **Maximizing reliability requires optimal neural diversity.** Across $P \in \{1, 2, 4, 8\}$ parallel representations and 6 hallucination benchmarks (182,850 samples, LOWESS, 80% CI), reliability follows an inverted-U, peaking at optimal $P_\star$ ($\Delta P = P - P_\star$) then degrading. Theorems 1 & 2 precisely predict the corresponding U-shaped hallucination probability ($R^2 = 0.943$, orange) and motivate principled architectural design (ND-LoRA), which reduces hallucinations 14.6% on average without degrading general capabilities.

| Category | Task | Best $P_\star$ | Best Score | $\Delta\%$ Score | Sig. |
|---|---|---|---|---|---|
| Hallucination | HaluEval (Dialog) | 4 | 0.516 | $+12.8\%$ | *** |
| | HaluEval (QA) | 4 | 0.451 | $+23.4\%$ | *** |
| | HaluEval (Summ) | 4 | 0.502 | $+25.6\%$ | *** |
| | MemoTrap v2 | 8 | 0.689 | $+8.8\%$ | *** |
| | TruthfulQA (MC1) | 2 | 0.269 | $+7.3\%$ | |
| | TruthfulQA (MC2) | 2 | 0.442 | $+9.5\%$ | * |
| Knowledge | NQ (8-shot) | 1 | 0.066 | – | |
| | NQ-swap | 8 | 0.554 | $+0.8\%$ | |
| | PopQA | 1 | 0.111 | – | |
| | TriviaQA (8-shot) | 1 | 0.192 | – | |

Table 1: **Optimal neural diversity is task-dependent: hallucination tasks benefit from neural diversity, knowledge tasks do not.** De-aggregating Figure 1, hallucination benchmarks consistently show large gains with increased diversity (up to 25.6%, HaluEval-Summ, $P_\star = 4$), while knowledge retrieval mostly peaks at $P_\star = 1$. This asymmetry supports hallucination as a reliability problem distinct from factual recall. Significance: $*** p < 0.001$, $* p < 0.05$.

# 1 Introduction

Despite scaling to trillions of parameters, language models hallucinate persistently (Lin et al., 2021). This reliability crisis is acute for small language models — increasingly favored for edge and agentic use cases (Zheng et al., 2025; Belcak et al., 2025) — whose compressed representations make them especially vulnerable to hallucinations, with even well-resourced efforts like GPT-OSS 20B exhibiting 91% hallucination rates on factual benchmarks (OpenAI, 2025).

Current hallucination mitigation strategies are largely empirically driven but theoretically ungrounded and target average performance rather than tail risk. RLHF optimizes mean harmlessness (Bai et al., 2022), RAG improves average factual grounding (Niu et al., 2024), and contrastive decoding enhances mean generation quality (Li et al., 2023b). While inference-time approaches like self-consistency and LoRA ensembling (Wang et al., 2022; 2023) reduce hallucinations through diverse sampling, they lack formal tail-probability guarantees. Similarly, parallel scaling methods (Chen et al., 2025) target first-moment improvements in perplexity and task accuracy. Yet controlling catastrophic failures requires bounding the tails of $\mathbb{P}(\text{hallucination})$, not just optimizing mean behavior.

Formal ensemble theory exists but targets the wrong objective. Classical ensemble methods (Krogh & Vedelsby, 1994) provide rigorous diversity theory to reduce mean generalization error $\mathbb{E}[\text{loss}]$, not tail-probability bounds for hallucinations. Deep ensembles (Lakshminarayanan et al., 2017) quantify uncertainty but lack hallucination-specific guarantees. Without explicit diversification, parallel architectures suffer *representational collapse* (Jing et al., 2022), leaving reliability gains unrealized.

Our key insight is that hallucinations — to the extent they arise from correlated representational errors rather than missing knowledge — are a form of noise amenable to portfolio-theoretic diversification. We demonstrate that a significant subset of hallucinations are in fact empirically addressable and theoretically tail-boundable by such diversification.

To our knowledge, we provide the first formal framework for **hallucination probability tail bounds in ensembled language models**, reframing it as a second-moment reliability problem. Drawing on portfolio theory (Markowitz, 1952), we prove that decorrelated parallel representations (*neural diversity*) reduce this tail bound and introduce **ND-LoRA (Neural Diversity Low-Rank Adaptation)**[1] to concretely demonstrate its hallucination reduction capabilities.

Our contributions are:

---

[1]Code, training and evaluation scripts, and checkpoints: `https://github.com/kushalc/nd-lora`.

- **Theoretical Linkage**: We reframe hallucinations as a second-moment reliability problem and prove (i) a portfolio-theoretic bound showing hallucination probability $\mathbb{P}(H) \propto 1/P$ with $P$ decorrelated parallel representations (Theorem 1); and, (ii) non-monotonicity in reliability scaling (Theorem 2), showing that excessive parallelism can degrade diversity (and thus reliability) under common circumstances. We further show (iii) our theoretical predictions achieve $R^2 = 0.943$ in fitting empirical reliability gains (Figure 1), establishing quantitative validation rare in neural hallucination research.

- **Constructive Demonstration**: We demonstrate empirical feasibility via ND-LoRA (parallel LoRA + Barlow Twins decorrelation), reducing hallucinations by up to 25.6% (and 14.6% on average) at $1.00008\times$ continued pretraining cost ($1.1\times$ inference latency) while preserving general capabilities across 12 tasks on a small language model (Qwen2.5-0.5B) (Table 1, 2).

- **Mechanistic Analysis**: We establish that neural diversity mediates hallucination in four ways: (i) causality via perturbation ($p < 0.001$, Table 4), (ii) quantitative scale via correlation (+0.1% diversity $\Leftrightarrow$ -3.8% hallucination, Figure 3), (iii) super-linear effects via ablation (Table 5), and (iv) task-dependent optima via scaling sweeps (Table 1).

Neural diversity enables reliability gains without scaling compute. While traditional scaling asks "how big?" and data scaling "how much?", diversity asks "how different?" — controlling tail probability through variance and correlation structure rather than capacity. Because the backbone stays frozen and only the parallel adapters train, this costs +0.008% of continued-pretraining compute; the sole recurring tradeoff is a $1.1\times$ inference latency from the $P$ parallel streams (subsection A.2).

## 2 A Theory of Neural Diversity

Why don't existing scaling methods improve reliability? Without explicit diversity mechanisms, gradient descent drives parallel streams toward similar representations through *representational collapse* (Jing et al., 2022), leaving reliability gains unrealized. We establish the first hallucination tail bounds for ensembled language models, proving that neural diversity reduces hallucinations and providing mathematical foundations for ND-LoRA.

Our strategy adapts portfolio theory to neural architecture design. Classical ensemble methods reduce *mean error* $\mathbb{E}[\text{loss}]$ through variance reduction (Krogh & Vedelsby, 1994), treating correlation as a factor that limits accuracy gains. In contrast, portfolio theory manages *tail risk* — rare but catastrophic failures — by diversifying across correlated assets (Markowitz, 1952). We adapt the latter framework to tail bound hallucination probability $\mathbb{P}(\text{hallucination})$, where correlation becomes the primary control variable for reliability rather than a secondary constraint on mean performance. We borrow terminology and analytical tools from financial econometrics throughout; subsection A.12 provides a glossary of non-standard deep learning and econometric terms for readers unfamiliar with either field.

### 2.1 Preliminaries

Modern language models hallucinate by fabricating facts, generating content inconsistent with input, or creating unsupported claims (Maynez et al., 2020; Ji et al., 2023). While comprehensive taxonomies exist (Huang et al., 2024), a useful factoring separates *faithfulness* (consistency with context) from *factuality* (correctness of recall). We primarily model the former via a simple signal-noise proxy that captures the underlying reliability failure — correlated representational errors — while remaining analytically tractable. However, as the two failure modes are not mutually exclusive, our technique can also improve factuality where correlated errors (not lack of knowledge) are the limiting factor.

**Signal-noise model.** Let $x \in X$ be a query with oracle output $y_\star(x) \in \mathbb{R}^V$ for vocabulary size $V$ and corresponding hidden representation $z_\star(x) \in \mathbb{R}^d$ for hidden dimension $d$. Consider an architecture that employs $P$ parallel computational pathways called *streams*, each processing the same input $x$ through the same model but in perturbed ways. We model the hidden output of each stream as $Z_i = z_\star + \varepsilon_i$ where $\varepsilon_i \in \mathbb{R}^d$ is centered noise with variance $\sigma_i^2 > 0$. The noise covariance $\Sigma \in \mathbb{R}^{P \times P}$ has entries $\Sigma_{ij} \triangleq \mathbb{E}[\langle \varepsilon_i, \varepsilon_j \rangle]$ with

pairwise correlations $\rho_{ij} \triangleq \Sigma_{ij}/(\sigma_i \sigma_j)$ for $i \neq j$. We aggregate hidden representations via $\widehat{Z}_w = \sum_i w_i Z_i$ with non-negative weights summing to one ($w_i \geq 0$, $\sum_i w_i = 1$). For readability, we omit $x$ where obvious and denote the average noise variance by $\bar{\sigma}^2 \triangleq \mathbb{E}[\sigma_i^2]$ and average correlation $\bar{\rho} \triangleq \mathbb{E}_{i<j}[\rho_{ij}]$.

**High-dimensional structure.** High-dimensional representations exhibit predictable geometric regularity that we exploit for analysis. We assume: (i) *Lipschitz decoding*, where outputs $\widehat{Y}_w(x) = f(\widehat{Z}_w(x))$ and $y_\star(x) = f(z_\star(x))$ satisfy $\|f(z) - f(z')\|_2 \leq L\|z - z'\|_2$ for some $L > 0$; (ii) *norm concentration*, where $\|\tilde{z}_i(x)\|_2^2 \approx d$ with small relative variance for per-feature whitened representations $\tilde{z}_i$; and, (iii) *feature alignment*, where in the readout eigenbasis $(e_k^\top \tilde{z}_i)(e_k^\top \tilde{z}_j) \geq 0$ for all features $k$ and stream pairs $(i, j)$ — i.e. features are self-consistent across streams and maintain polarity. Properties (i) and (ii) are standard in high-dimensional probability (Vershynin, 2018) and neural network analysis (Fazlyab et al., 2019; Bartlett et al., 2017). Property (iii) holds naturally in parallel architectures: streams share a backbone and combine via convex aggregation, so each stream's projection onto a readout eigenvector inherits the same sign from the shared backbone; Table 5 confirms this empirically (neural diversity index $\mathcal{D} \sim 1$; see below) for standard parallel architectures.

**Neural representations.** At a chosen design layer, each stream $i$ exposes a $d$-dimensional hidden representation $z_i(X)$. We whiten per-feature to obtain $\tilde{z}_i$ with zero mean and identity covariance. For streams $i < j$, the cross-correlation matrix is $C^{(ij)} \triangleq \mathbb{E}\left[\tilde{z}_i \tilde{z}_j^\top\right] \in \mathbb{R}^{d \times d}$ whose diagonal entries measure same-feature similarity and off-diagonal entries capture cross-feature alignment. Finally, using the widely-exploited observation that trained networks exhibit locally linear behavior at their operating point (Goodfellow et al., 2015; Simonyan et al., 2014), we connect representations to noise via local linearity: $\xi_i = A\tilde{z}_i$ for a shared linear readout $A \in \mathbb{R}^{V \times d}$ with finite condition number $\kappa = s_{\max}/s_{\min}$ over its $\min(V, d) = d$ singular values.

**Neural diversity index.** We define a simple cosine-based index to measure cross-stream diversity:

$$\mathcal{D} \triangleq \sqrt{\mathbb{E}_{i<j}\left[\frac{(\tilde{z}_i \cdot \tilde{z}_j)^2}{\|\tilde{z}_i\|^2 \|\tilde{z}_j\|^2}\right]}. \tag{1}$$

Lower $\mathcal{D}$ indicates greater neural diversity: $\mathcal{D} = 0$ means all streams are perfectly orthogonal, while $\mathcal{D} = 1$ means streams have suffered complete collapse.

**Hallucinations.** We define the output error as $E_w \triangleq \|\widehat{Y}_w(x) - y_\star(x)\|_F$, which is comparable to metrics like TruthfulQA-MC2 (Lin et al., 2021). For tolerance $\delta > 0$, the *hallucination event* is $H_\delta \triangleq \{E_w \geq \delta\}$. Our goal is to bound $\mathbb{P}(H_\delta)$ as a function of neural diversity $\mathcal{D}$ across streams $P$.

## 2.2 Neural Diversity Bounds Hallucination

Classical portfolio theory (Markowitz, 1952) gives the variance of an equally weighted portfolio of $P$ assets with average variance $\bar{\sigma}^2$ and average pairwise correlation $\bar{\rho}$ as:

$$\text{Var}(Y) = \bar{\sigma}^2\left(\frac{1 - \bar{\rho}}{P} + \bar{\rho}\right). \tag{2}$$

To use this observation for hallucinations, we must first connect neuron-level representations to portfolio-level noise correlations. Exploiting the fact that (i) our ensemble has one underlying model with aligned neuron-level representations and (ii) our model has geometric regularity in representation and output, the following lemma establishes this mapping:

**Lemma 1** (Average Correlation Bound). *Suppose there exists a kurtosis bound $C_4 \geq 1$ such that $\mathbb{E}[\|\xi_i\|_2^4] \leq C_4\,\sigma_i^4$ for all streams $i$. Then the average pairwise noise correlation $\bar{\rho}$ satisfies*

$$|\bar{\rho}| \leq C_* \mathcal{D}, \tag{3}$$

*where $C_* \triangleq \sqrt{C_4}\,\kappa^2$ depends on the kurtosis bound $C_4$ and readout condition number $\kappa$.*

*Proof sketch.* We proceed in two steps. Let $\mathcal{D}_\xi$ denote the diversity index computed over noise vectors $\xi_i$ (analogous to $\mathcal{D}$ but in readout space), with pairwise terms $\mathcal{D}_{\xi,ij}$. *(1)* Spectral bounds and feature alignment imply the linear readout distorts cosines by at most $\kappa^2$, so $\mathcal{D}_\xi \leq \kappa^2 \mathcal{D}$. *(2)* Cauchy–Schwarz twice — inner products to cosines, then kurtosis — gives $|\rho_{ij}| \leq \sqrt{C_4}\, \mathcal{D}_{\xi,ij}$; averaging pairs completes the proof. $\qquad\square$

We now have a direct path to tail-bound $P(\mathrm{H}_\delta)$ as a function of $\mathcal{D}$ and $P$. For readability, we assume uniform weights $w_i = 1/P$ below but our approach can also be easily applied to arbitrary weights.

**Theorem 1** (Hallucination Bound with Diversity). *For any tolerance $\delta > 0$, the hallucination probability $\mathbb{P}(\mathrm{H}_\delta)$ satisfies*

$$\mathbb{P}(\mathrm{H}_\delta) \ \leq \ \frac{\frac{1 - C_* \mathcal{D}}{P} + C_* \mathcal{D}}{\frac{1 - C_* \mathcal{D}}{P} + C_* \mathcal{D} + SNR}, \tag{4}$$

*where $SNR \triangleq \delta^2 / \bar{\sigma}^2$ is the signal-to-noise ratio, $\mathcal{D}$ is the neural diversity index (Equation 1), $C_* = \sqrt{C_4}\, \kappa^2$ is the readout-kurtosis constant (Lemma 1), and $P$ is the number of parallel streams as above.*

*Proof sketch.* Lemma 1 bounds $|\bar{\rho}| \leq C_* \mathcal{D}$, linking noise variance to representational diversity. Plugging into Equation 2, applying Chebyshev and normalizing by $\bar{\sigma}^2$ yields the stated bound. $\qquad\square$

This completes the first half of our theoretical result: Neural diversity mediates hallucination probability. With perfect de-correlation ($\bar{\rho} = 0$), hallucination probability scales as $O(1/P)$ — more streams reduce hallucination risk. When streams collapse ($\bar{\rho} = 1$), the bound becomes independent of $P$, explaining why naive ensembling without diversification provides no reliability benefits.

## 2.3 Non-Monotonic Scaling Behavior

Next, we demonstrate that under common circumstances, the hallucination bound follows a U-shaped curve — initially decreasing with higher $P$, but starts increasing eventually. Consider the case where the correlation itself increases with $P$, say, due to optimizer constraints:

**Theorem 2** (U-shaped Behavior). *Suppose $\bar{\rho}(P) = \rho_0 + \beta(P-1)^\gamma$ for constants $\rho_0 \in [0,1)$, $\beta > 0$, $\gamma > 0$. Define*

$$v(P) \triangleq \mathrm{Var}(E_w) = \bar{\sigma}^2 \left( \frac{1 - \bar{\rho}(P)}{P} + \bar{\rho}(P) \right), \qquad \mathcal{B}(P) \triangleq \frac{v(P)}{v(P) + \delta^2}. \tag{5}$$

*Then $\mathcal{B}(P)$ is U-shaped: there exists $P_\star \geq 1$ minimizing $\mathbb{P}(\mathrm{H}_\delta)$, with $P_\star$ controlled by how fast $\bar{\rho}(P)$ degrades with $P$.*

*Proof sketch.* The hallucination bound $\mathcal{B}(P)$ is monotonic in variance $v(P)$, so we analyze $v(P)$ directly. There are two competing effects: the $1/P$ term drives variance down, while growing correlation $\bar{\rho}(P) = \rho_0 + \beta(P-1)^\gamma$ eventually dominates. Differentiating shows $v'(P)$ changes sign exactly once, yielding a unique minimum $P_\star$ whose location depends $\beta, \gamma$ and $\rho_0$. $\qquad\square$

This theorem establishes *non-monotonicity* — hallucination probability $\mathbb{P}(\mathrm{H}_\delta)$ actually *increases* for larger $P$, meaning reliability degrades. This is stronger than the well-known diminishing returns of ensembles (where improvement slows but continues). While ensemble theory also shows optimal size matches the number of class labels for accuracy-optimized classifiers (Bonab & Can, 2019), we prove and validate (Figure 1) that diversity can degrade in generative language models with excessive parallelism under common circumstances and also harm reliability.

## 2.4 Theoretical Validation

By measuring empirical diversity $\mathcal{D}(P)$ and plugging these values into Theorem 1's bound, we achieve $R^2 = 0.943$ (Figure 1), explaining 94.3% of empirical reliability variation. This fit uses only two free parameters ($C_*$, $SNR$) shared across all tasks and observations, with $\mathcal{D}(P)$ fixed from empirical measurements. Theorem 2's correlation growth model provides a mechanism for observed concavity: correlation grows as

$O((P-1)^{\gamma})$, overwhelming the $O(1/P)$ diversification benefit. This alignment — rare in hallucination research where theory often lags empirics — validates our portfolio-theoretic framework.

Together, Theorem 1 and Theorem 2 show that (i) reducing $\mathcal{D}$ reduces hallucinations and (ii) there exists an optimal $P_{\star}$ that minimizes hallucinations. Next, we show how to construct an architecture and training protocol to reduce $\mathcal{D}$ and find $P_{\star}$.

## 3   ND-LoRA: A Practical Demonstration

### ND-LoRA Architecture (P=4)

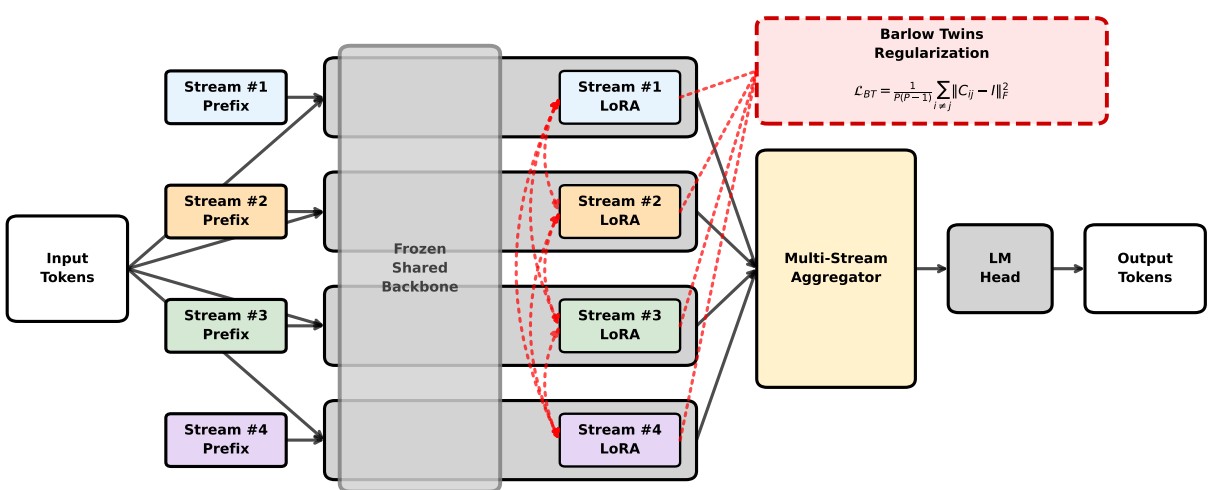

Figure 2: **ND-LoRA schematic for $P=4$ parallel streams.** Each stream receives independent LoRA adapters and learnable prefix tokens. The aggregator combines stream outputs with learnable weights, while Barlow Twins regularization incentivizes decorrelation between stream outputs.

We introduce ND-LoRA (Neural Diversity Low-Rank Adaptation), a parameter-efficient method that demonstrates our theoretical framework for neural diversity regularization. ND-LoRA extends the ParScale architecture with stream-aware LoRA adapters and explicit decorrelation objectives. Figure 2 visually summarizes our approach.

### 3.1   Architecture

Our implementation builds on ParScale with $P$ parallel computation streams. Each stream $i \in \{1, \ldots, P\}$ uses 48 learnable prefix tokens prepended to the input sequence that flow through all layers via the attention mechanism, along with stream-specific LoRA adapters applied at each layer:

$$h_i^{(\ell)} = \text{Layer}^{(\ell)}(h_i^{(\ell-1)} + B_i^{(\ell)} A_i^{(\ell)} h_i^{(\ell-1)}) \tag{6}$$

where $B_i^{(\ell)} \in \mathbb{R}^{d \times r}$, $A_i^{(\ell)} \in \mathbb{R}^{r \times d}$ are stream-specific LoRA matrices with rank $r$. The final output combines streams through a learned aggregator:

$$y = \text{LM\_Head}\left(\sum_{i=1}^{P} w_i \cdot h_i^{(L)}\right) \tag{7}$$

where $w_i = (1-\varepsilon) \cdot \text{softmax}(\text{MLP}([h_1^{(L)}, \ldots, h_P^{(L)}]))_i + \varepsilon/P$ are dynamic weights with label smoothing ($\varepsilon = 0.1$) computed from the concatenated stream representations. This prevents attention collapse by ensuring minimum weight $\varepsilon/P$ for each stream.

| Model | HaluEval | MemoTrap | TruthfulQA | NQ | Wikitext | WG |
|---|---|---|---|---|---|---|
| ND-LoRA R16 (P=2) | **0.481*** | **0.666*** | **0.442*** | 0.055 | 0.784 | **0.574** |
| ParScale R32 (P=2) | 0.439 | 0.638 | 0.412 | 0.059 | 0.793 | 0.564 |
| Qwen LoRA R32 | 0.400 | 0.634 | 0.403 | **0.065** | **0.778** | 0.572 |

Table 2: **Even at $P = 2$ streams, ND-LoRA achieves up to 20.2% relative hallucination reduction vs. parameter-matched baseline.** Across hallucination benchmarks, ND-LoRA shows statistically significant improvements (HaluEval-Summarization, MemoTrap, TruthfulQA-MC2) while maintaining competitive Winogrande, NQ, and Wikitext BPB (lower is better) general-purpose capabilities. Baselines use higher LoRA ranks for parameter parity. * indicates $p < 0.05$.

This architecture enables stream specialization while maintaining parameter efficiency. For $P = 2$ streams with rank-16 LoRA, we use approximately 29K trainable parameters per layer, comparable to a single rank-32 LoRA but with fundamentally different representational capabilities.

Concretely, this architecture approximates Lemma 1's shared-readout simplification. The post-design computation factors as a frozen backbone (layers $\ell_\star+1, \ldots, L$) composed with a single lm_head, both shared across streams. Only rank-16 LoRA adapters perturb this post-design readout (pre-design adapters shape $\tilde{z}_i$ itself), giving $A_i = A + \Delta A_i$ with $\text{rank}(\Delta A_i) \leq 16 \ll \min(V, d) = d = 896$, so $A_i \approx A$. The $R^2 = 0.943$ fit in Figure 1 confirms that the theory (including modeling simplifications like shared-readout, linearization and norm concentration) bridges well to practice.

### 3.2 Barlow Twins Regularization

To encourage neural diversity, we apply Barlow Twins regularization across all pairs of streams $i < j$ at a pre-specified design layer $\ell_\star$.

Let $z_i \in \mathbb{R}^{B \times T \times d}$ denote the hidden representations of stream $i$ at the design layer for a batch of size $B$ and sequence length $T$. We first apply batch normalization and mean-centering to obtain whitened features $\tilde{z}_i$. We then calculate the cross-correlation matrices $C^{(ij)} \in \mathbb{R}^{d \times d}$ as in subsection 2.2 and apply standard Barlow Twins (Zbontar et al., 2021) for each pair of streams $i < j$:

$$\mathcal{L}_{BT} = \mathbb{E}_{i<j} \left\| C^{(ij)} - I \right\|_F \tag{8}$$

The total training objective combines cross-entropy and decorrelation terms:

$$\mathcal{L} = \mathcal{L}_{CE} + \lambda_{BT} \mathcal{L}_{BT} \tag{9}$$

Notably, driving $C^{(ij)} \to I$ reinforces the feature alignment assumption (subsection 2.1): diagonal entries stay positive while off-diagonal leakage is suppressed.

## 4 Experimental Validation

We validate ND-LoRA through systematic hallucination reduction experiments using parameter- and data-matched comparisons. Although all our theoretical analysis (section 2) is model-agnostic, all empirical results in the following sections are on a single small language model (Qwen2.5-0.5B). We describe our full experimental setup in subsection A.4.

### 4.1 Key Results

Table 2 demonstrates ND-LoRA achieves substantial improvements on hallucination-sensitive benchmarks while maintaining competitive general performance. ND-LoRA with $P = 2$ streams achieves statistically significant improvements on HaluEval-Summarization (0.481* vs 0.400, $p < 0.001$, 8.1% absolute / 20.2%

| Method | Type | Halluc. $\Delta\%$ | Knowledge $\Delta\%$ |
|---|---|---|---|
| ND-LoRA | integrated | **+14.6%** | +0.2% |
| CAD | inference-time | +4.1% | **+1.2%** |
| ActDec | inference-time | +1.5% | -2.6% |
| Disagreement | training-time | +1.7% | -1.1% |

Table 3: **ND-LoRA dominates on hallucination without a knowledge tax.** Average relative $\Delta\%$ vs. the $P = 1$ baseline across six hallucination (HaluEval Dial/QA/Summ, MemoTrap, TF-MC1/MC2) and five knowledge (NQ, PopQA, TriviaQA, Winogrande, Wikitext BPB) benchmarks.

relative), TruthfulQA-MC2 (0.442* vs 0.403, $p = 0.030$, 3.9% absolute / 9.5% relative) and MemoTrap (0.666* vs 0.634, $p < 0.001$, 3.2% absolute / 5.1% relative) vs parameter-matched Qwen, validating our theoretical prediction.

Although ND-LoRA's improvements specifically target reliability benchmarks, they preserve general capabilities. Qwen slightly outperforms on Wikitext (0.778 vs. 0.784) and Natural Questions (0.065 vs. 0.055), but ND-LoRA wins slightly on Winogrande (0.574 vs. 0.572).

Parameter efficiency is evident comparing ND-LoRA R16 ($P = 2$) against Qwen2.5-0.5B LoRA R32. Despite lower-rank adapters, ND-LoRA consistently outperforms the high-rank baseline on hallucination tasks, demonstrating that architectural diversity provides more value at equal capacity. This shows representational diversity, not parameter count, drives reliability gains in our experiment.

These findings establish neural diversity as a practical reliability mechanism. Consistent improvements across hallucination benchmarks with preserved general performance suggest ND-LoRA addresses fundamental reliability challenges rather than metric-specific optimization. Figure 3 demonstrates strong empirical correlation between neural diversity and performance, building intuition for the causal relationship established in subsection 5.1.

## 4.2 Task-Dependent Optimality

Further, the optimal diversity is task-dependent. Table 1 reveals striking task-dependent sensitivity patterns relative to the $P = 1$ baseline. Hallucination-focused tasks show the largest gains: HaluEval Summarization achieves +25.6% relative improvement at $P = 4$, HaluEval QA shows +23.4% at $P = 4$, and TruthfulQA MC2 shows +9.5% at $P = 2$ while MemoTrap benefits from higher diversity ($P = 8$, +8.8%). Notably, knowledge-intensive tasks like PopQA, TriviaQA and NQ show no improvement over baseline, which is expected as ND-LoRA does not add new sources of knowledge or try to improve recall of existing knowledge. This heterogeneity demonstrates that different tasks require different amounts of neural diversity to maximize reliability, with hallucination-focused tasks generally benefiting most from decorrelated representations.

Disaggregating further into *faithfulness* tasks (HaluEval-Dialog, -QA, -Summarization, MemoTrap v2) and *factuality* tasks (TruthfulQA-MC1, -MC2) reveals a consistent $\approx 2\times$ advantage for faithfulness (subsection A.7). This is theoretically expected: neural diversity decorrelates the internal verification of context-grounded claims, directly benefiting faithfulness. In contrast, factuality requires knowledge the model may lack, which diversity alone cannot supply, limiting diversity-based gains.

## 4.3 Comparison with Other Methods

Table 3 compares ND-LoRA to three baselines on Qwen2.5-0.5B: Context-Aware Decoding (CAD; Shi et al., 2024) and Activation Decoding (ActDec; Chen et al., 2024) are inference-time interventions; Disagreement Regularization (Li et al., 2018) is a compute-matched training-time method. ND-LoRA yields +14.6% hallucination improvement — 3.5× the next-best baseline (CAD, +4.1%) — while keeping knowledge within 0.2% of the $P = 1$ baseline (vs. CAD +1.2%, ActDec −2.6%, Disagreement −1.1%). Per-benchmark scores (subsection A.6) show ND-LoRA winning every hallucination column; CAD edges out PopQA and TriviaQA by 1–5 basis points.

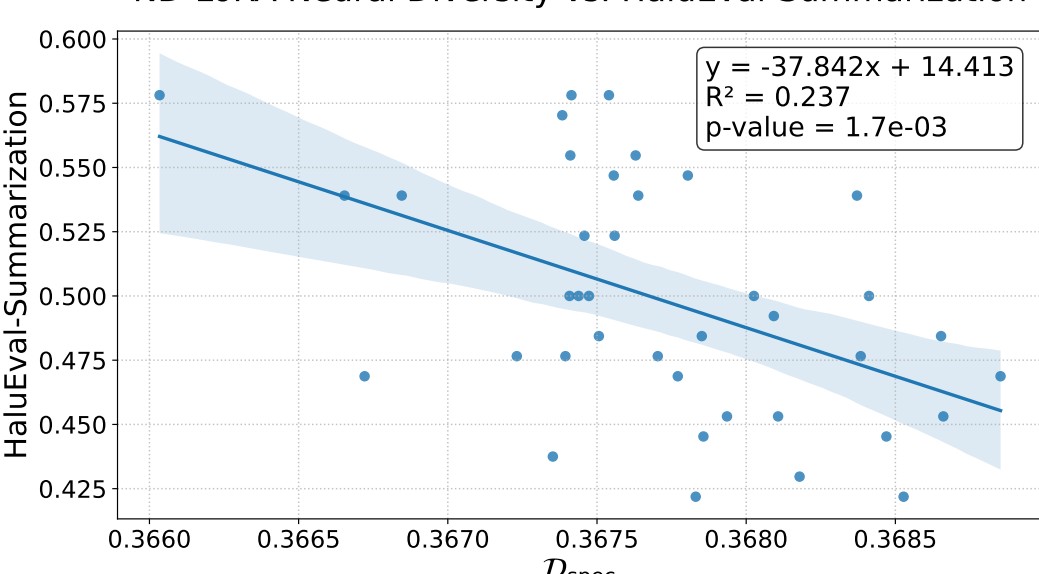

Figure 3: **Reliability improves as neural diversity increases (lower $\mathcal{D}$).** Specifically, diversity ($\mathcal{D}$) is negatively correlated with HaluEval-Summarization performance (slope=-37.842, R²=0.237, p=0.002), consistent with $\mathbb{P}(H) \propto \mathcal{D}$ in Theorem 1.

| Task | $\Delta\mathcal{D}$ | $\Delta$ Score | SE | d | p-value | Sig. | N |
|------|------|------|------|------|------|------|------|
| HaluEval-Summ | 0.024 | -0.005 | 0.010 | 0.007 | $1.6 \times 10^{-5}$ | *** | 512 |
| MemoTrap v2 | 0.031 | -0.003 | 0.010 | 0.000 | $8.2 \times 10^{-5}$ | *** | 512 |
| TruthfulQA-MC2 | 0.025 | -0.007 | 0.009 | 0.018 | $3.3 \times 10^{-7}$ | *** | 512 |

Table 4: **Artificial corruption of neural diversity establishes statistical causality.** Perturbing neural diversity ($\Delta\mathcal{D} > 0$) causes accuracy drops across tasks with high statistical significance ($p < 0.001$) via paired t-tests with Fisher meta-analysis (N=4 sub-experiments $\times$ 128 samples each).

## 5 Mechanistic Analysis

### 5.1 Neural Diversity as the Causal Mediator

To establish causality beyond correlation, we perform artificial corruption interventions that directly manipulate cross-stream similarity.

**Experiment Design.** Starting with a pre-trained ND-LoRA $P = 4$ model, we inject a corruption hook at the RMSNorm layer that randomly substitutes the hidden state at randomly-chosen positions in a given stream from another stream, perturbing $\mathcal{D}$ while preserving activation magnitudes. We evaluate on a matched basis: each corrupted evaluation is paired with an uncorrupted baseline using identical samples and resampling indices. Across 4 sub-experiments with different random seeds, we collect $N = 128$ paired samples per task. This paired design maximizes statistical power by controlling sample-level variance, analyzed via paired t-tests with Fisher meta-analysis.

**Results.** Table 4 provides statistically robust evidence that neural diversity causally affects performance. All three tasks show highly significant accuracy drops ($p < 0.001$) when stream-level substitution perturbs diversity ($\Delta\mathcal{D} \approx 0.025$). While effect sizes are modest (0.3% to 0.7% score reduction) — likely because artificial stream substitution creates out-of-distribution corruption patterns — the statistical significance establishes causality beyond correlational association.

## 5.2 Ablations

To isolate the contributions of ND-LoRA, we systematically ablate ND-LoRA components at fixed $P = 4$ streams. All variants maintain parameter parity through LoRA rank adjustments, enabling fair comparison. We measure inference-time diversity ($\mathcal{D}$) at the aggregation layer using evaluation samples, quantifying actual cross-stream correlation during inference.

Table 5 reveals a super-linear combination: independent LoRA (+2.9%) and Barlow Twins (+1.4%) sum to 4.3% but achieve 4.9% when combined (Stream LoRA-BT) — a 14% bonus. Targeting KVQ attention amplifies this further by 2.6× to +12.8% (ND-LoRA at fixed $P = 4$; maximum gains reach 14.6% when optimizing $P$ per-task, see Table 1). Neither component alone suffices: ParScale's near-complete collapse ($\mathcal{D} = 0.9990$) yields only +0.5%, while Stream LoRA without regularization achieves +2.9%, both less than a quarter of ND-LoRA's final impact. This establishes that both architectural capacity and explicit regularization are necessary for full impact.

Notably, ParScale's original work found prefix tuning superior to LoRA for mean loss (Table 6 in Chen et al. 2025). However, stream-aware LoRA is necessary for reducing tail probability: even with Barlow Twins, prefix tuning collapses streams ($\mathcal{D} = 0.9988$), while stream-aware LoRA enables decorrelation ($\mathcal{D} = 0.1530$). This illustrates how second-moment objectives require different architectural choices than first-moment objectives.

Counterintuitively, ND-LoRA achieves best performance (+12.8%) with *higher* $\mathcal{D} = 0.4112$ than Stream LoRA-BT's 0.1530. This reveals that strategic localization to representational bottlenecks matters more than maximizing global decorrelation: focusing LoRA and Barlow Twins on KVQ attention modules provides 2.6× amplification. This further reinforces how second-moment objectives differ architecturally from first-moment ones and, consistent with Table 1, that neural diversity is a task-dependent resource requiring strategic allocation to critical computational pathways.

## 5.3 Practical Approximability

While task-optimal $P_\star$ varies (Table 1), practitioners need not search exhaustively. Defaulting to $P = 4$ achieves 97% of oracle hallucination performance (96% across all 12 evaluations). Additionally, a simple router (subsection A.9) achieves 99% of oracle hallucination performance by predicting $P$ from prompt statistics, revealing a retrieval-vs-verifiability tradeoff: question-dense prompts favor low $P$, while longer, context-heavy prompts favor higher $P$.

## 5.4 Hyperparameter Sensitivity

Sensitivity analyses across 70+ configurations show that hallucination improvements are stable across design layers $\ell_\star \in [7, 23]$ while $\lambda_{BT} \in [0.01, 0.50]$ exposes a hallucination–perplexity tradeoff (subsection A.11); LoRA rank $R16$–$R128$ and alpha scalings are not confounds and attention-only LoRA outperforms MLP-only LoRA (subsection A.10); and even worst-choice $P \in \{2, 4, 8\}$ improves hallucination over the parameter-matched baseline (subsection A.8).

## 5.5 Computational Considerations

Unlike $P$-model ensembles with $P\times$ pretraining cost, ND-LoRA achieves substantial reliability gains at negligible overhead (1.00008× pretraining, 1.1× latency) given its single architecture. Parallelized 20M amortizes to ≈0.008% of 1T-token pretraining, frozen backbone makes gradients nearly free, and ND-LoRA requires near-identical FLOPs to ParScale at inference, with per-stream adapters servable through a batched adapter kernel (Chen et al., 2023). See subsection A.2.

# 6 Related Work

**Hallucination in Language Models.** Hallucinations represent a fundamental challenge in modern language models. Comprehensive surveys establish taxonomies that distinguish factuality vs. faithfulness

| Variant | Streams | LoRA | Regul. | Target | $\mathcal{D}$ | $\overline{\Delta}$% Score | $\Delta$ Cost |
|---|---|---|---|---|---|---|---|
| Standard | 1 | Single | D | All | – | 0.0% | **1.0x / 1.0x** |
| ParScale | $P$ | Single | D | All | 0.9990 | +0.5% | 1.00008x / 1.1x |
| ParScale-BT | $P$ | Single | D + BT | All | 0.9988 | +1.4% | 1.00008x / 1.1x |
| Stream LoRA | $P$ | Stream | D | All | 0.3544 | +2.9% | 1.00008x / 1.1x |
| Stream LoRA-BT | $P$ | Stream | D + BT | All | **0.1530** | +4.9% | 1.00008x / 1.1x |
| ND-LoRA | $P$ | Stream | D + BT | KVQ | 0.4112 | **+12.8%** | 1.00008x / 1.1x |

Table 5: **Ablations reveal super-linear combination of impact.** Stream LoRA (+2.9%) and Barlow Twins (+1.4%) combine super-linearly (+4.9%), and focusing on KVQ attention amplifies to +12.8%. *LoRA*: single shared vs. $P$ stream-aware adapters. *Regularization*: Dropout vs. Barlow Twins. *Target*: All layers vs. KVQ attention only. $\mathcal{D}$: Neural Diversity Index (lower is better). $\overline{\Delta}$% *Score*: avg. change (hallucination benchmarks). Ablations shown at fixed $P = 4$ streams.

(Huang et al., 2024; Tonmoy et al., 2024). Theoretical work proves hallucinations are mathematically inevitable in computable models under certain resource constraints (Xu et al., 2024; Kalai & Vempala, 2024), with smaller models exhibiting particular severity on factual benchmarks (Lin et al., 2021; Li et al., 2023a). Mechanistic investigations reveal hallucinations arise from internal representation failures (Yu et al., 2024), knowledge awareness limitations (Ferrando et al., 2025), and attention pattern anomalies.

Mitigation has predominantly targeted average performance. Retrieval augmentation (RAG) incorporates external knowledge for factual grounding (Niu et al., 2024). RLHF improves alignment (Bai et al., 2022), while constitutional AI enhances safety. Decoding methods use contrastive decoding (Li et al., 2023b) and classifier-free guidance (Sanchez et al., 2023). Critically, improving $\mathbb{E}[\text{error}]$ does not guarantee improvements to $\mathbb{P}(\text{hallucination})$, as tail events depend on variance and correlation structure, not just central tendency.

Second-moment approaches exist but lack theoretical grounding: self-consistency reduces hallucinations through diverse sampling (Wang et al., 2022) without formal tail-probability guarantees, while deep ensembles provide uncertainty estimates (Lakshminarayanan et al., 2017) but not hallucination-specific bounds. We provide the first formal tail bounds connecting neural diversity to hallucination probability as a second-moment problem.

**Deep Ensembles, Parallel Architectures & Inference-Time Scaling** Deep ensembles provide uncertainty estimates (Lakshminarayanan et al., 2017) with power-law scaling (Lobacheva et al., 2020) for calibration and OOD detection. LLM ensembles benefit from explicit diversity optimization (Tekin et al., 2024), while negative correlation learning demonstrates diversity must be actively encouraged (Liu & Yao, 1999). The "memory split advantage" shows ensembles of smaller models can outperform single large models at fixed parameter budgets. Optimal size theory reveals weighted voting exhibits diminishing returns due to correlation and overfitting (Bonab & Can, 2019), with predictions stabilizing at 5–10 models (Hernández-Lobato et al., 2013). These approaches require multiple independent models, incurring $P\times$ training and inference costs.

Inference-time methods reduce hallucinations by reshaping the decoding distribution or diversifying over sampling noise. Context-Aware Decoding (CAD; Shi et al., 2024) and Activation Decoding (ActDec; Chen et al., 2024) steer logits toward context-grounded generations; we benchmark ND-LoRA head-to-head against both in subsection 4.3. Self-consistency (Wang et al., 2022), confidence-based weighting (Taubenfeld et al., 2025), and contrastive decoding (Li et al., 2023b) instead aggregate over multiple generations from a fixed representation. All of these intervene only at inference — leaving the underlying representations unchanged — whereas our training-time parallelism learns coordinated streams; because the two attack at different stages of the pipeline (representation learning vs. decoding), they compose rather than substitute.

Self-ensembled parallel architectures like ParScale (Chen et al., 2025) break the multiplicative memory requirements of classical ensembles by using $P$ perturbed computational pathways within a single model. ParScale achieves $O(\log P)$ general capability gains, modeling parallel streams with correlation $\rho$ in scaling laws $L \propto (N \cdot P^{1/\alpha} \cdot [(P-1)\rho+1]^{-1/\alpha})^{-\alpha}$. This targets mean loss for accuracy improvements, not hallucination

probability. We directly build our demonstration on ParScale, extending their theoretical framework and implementation to tail-bound hallucinations.

**Theoretical Foundations.** Modern portfolio theory (Markowitz, 1952) provides the mathematical foundation for understanding correlation-based risk reduction, with diversification principles (Meucci, 2009) for ensemble variance analysis. Classical ensemble theory reduces mean error $\mathbb{E}[\text{loss}]$ via variance decomposition (Dietterich, 2000). PAC-Bayesian bounds connect diversity to minimax-optimal generalization (Ortega et al., 2022) and concentration inequalities showing correlation reduction tightens tail bounds (Alquier, 2024). We link these frameworks to modern neural networks to bound hallucination tail probabilities.

**Redundancy Reduction.** A rich history of diversification exists in self-supervised learning to avoid training collapse and in PEFT methods for efficient specialization. Self-supervised approaches like Barlow Twins (Zbontar et al., 2021) and VICReg (Bardes et al., 2022) use decorrelation to prevent dimensional collapse (Jing et al., 2022). PEFT methods like LoRA (Hu et al., 2022) and prefix-tuning (Li & Liang, 2021) enable model specialization under limited parameter budgets, with BatchEnsemble and LoRA-Ensemble achieving diversity through parameterization (Wen et al., 2020; Mühlematter et al., 2025). We adapt these methods for second-moment reliability guarantees.

## 7 Discussion

At a time when the reliability of language models is becoming the critical barrier to real-world deployment, we (i) provide the first formal framework to tail-bound hallucinations in ensembled language models, demonstrating that neural diversity plays a critical role in reducing hallucinations; and, (ii) using this technique, achieve up to 25.6% (and 14.6% on average) reduction in hallucination rates on evaluated benchmarks at fixed parameter and data budgets at +0.008% pretraining cost. Neural diversity enables reliability gains without massive compute scaling; the only recurring cost is a $1.1\times$ inference latency from evaluating $P$ parallel streams (subsection A.2), a modest tradeoff for the reliability gains in latency-tolerant, safety-critical deployments.

By reframing hallucinations as a second-moment problem — controlled through variance and correlation rather than mean optimization — we open an under-explored research direction orthogonal to existing approaches. While RLHF and RAG target first-moment improvements (average performance), neural diversity targets tail probability through explicit decorrelation. This bridges portfolio theory to neural reliability, a connection previously unexplored. The gap between extensive first-moment research and nascent second-moment approaches (self-consistency, our work) suggests substantial opportunity for reliability-focused methods grounded in tail-probability theory.

Our small-scale demonstration and mechanistic analysis validates the theoretical framework; scaling to larger models is a natural next step given that continued training requires only +0.008% additional overhead and $P = 4$ captures 94.3% of oracle performance. The task-dependent optimal $P_\star$ in Table 1 reveals intriguing structure, suggesting deeper connections between task complexity, knowledge recall vs. precision and neural diversity worthy of theoretical characterization.

Although the task-dependent optimal $P$ introduces additional complexity via a new hyperparameter, three observations mitigate this concern. First, even at worst-case $P$, ND-LoRA still beats the parameter-matched baseline on every hallucination benchmark (subsection A.8). Second, defaulting to $P=4$ captures 97% of oracle hallucination performance with no tuning. Third, a simple two-feature router achieves 99% of oracle hallucination performance (subsection A.9), with learned coefficients reinforcing knowledge recall vs. precision dynamics above (subsection A.7).

Our work opens three immediate research directions: (i) *Theoretical*: characterizing optimal $P_\star$ as a function of task properties via information-theoretic approaches suggested by the U-shape theorem (Theorem 2). (ii) *Practical*: combining neural diversity with inference-time scaling (Snell et al., 2024) for multiplicative reliability gains. (iii) *Extension*: preliminary work suggests portfolio-theoretic dynamics emerge in single-stream language models via internal parallel structures (e.g. multi-head attention), but co-exist with interference dynamics requiring a distinct theoretical treatment. Second-moment reliability is an essential frontier as language models become critical infrastructure in high-stakes domains.

**Author Contributions**

Kushal Chakrabarti led the overall project design, core implementation of ND-LoRA, experimental design, and paper writing. Nirmal Balachundhar contributed theoretical insights and implementation for novel neural diversity techniques, experiment implementation and analysis, and paper writing.

**Acknowledgments**

We thank the research community at South Park Commons for valuable discussions and feedback throughout this project. We are grateful to Augustine Mavor-Parker, Bhav Ashok, Daniel Morillo, David Sontag, Iman Modarressi, Jaclyn Lunger, Javier Ferrando, John Bohannon, Nelson Ray, Patrick Bozeman and Suhith Rajesh for their insightful comments on earlier drafts. This work was supported by South Park Commons and Modal, who generously provided compute resources for our experiments.

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

# A  Appendix

## A.1  Full Proofs

### A.1.1  Proof of Lemma 1

*Proof.* We proceed in two steps. First, we show that the shared linear readout $A$ can at most distort cosines between representations by a factor of $\kappa^2$. Second, we convert bounded cosine alignment between the error vectors $\xi_i$ into a bound on average noise correlation.

**Step 1: local linear readout and cosine distortion.** Let $s_{\min}$ and $s_{\max}$ denote the minimal and maximal singular values of $A$ and $\kappa = s_{\max}/s_{\min}$ its condition number. For streams $i, j$ with representations $\tilde{z}_i, \tilde{z}_j \in \mathbb{R}^d$, the local linear readout gives $\xi_i = A\tilde{z}_i$ and $\xi_j = A\tilde{z}_j$. Then

$$\cos\angle(\xi_i, \xi_j) \;=\; \frac{\langle \xi_i, \xi_j \rangle}{\|\xi_i\|_2 \,\|\xi_j\|_2} \;=\; \frac{\tilde{z}_i^\top A^\top A \,\tilde{z}_j}{\|A\tilde{z}_i\|_2 \,\|A\tilde{z}_j\|_2}.$$

Write $\tilde{z}_i^\top A^\top A\,\tilde{z}_j = \sum_k s_k^2 a_k b_k$ where $a_k = e_k^\top \tilde{z}_i$, $b_k = e_k^\top \tilde{z}_j$, $s_k$ are the singular values of $A$, and $\{e_k\}$ are its right singular vectors (equivalently, eigenvectors of $A^\top A$ with eigenvalues $s_k^2$). By feature alignment (subsection 2.1), $a_k b_k \geq 0$ for all $k$, so:

$$|\tilde{z}_i^\top A^\top A\,\tilde{z}_j| \;=\; \sum_k s_k^2 a_k b_k \;\leq\; s_{\max}^2 \sum_k a_k b_k \;=\; s_{\max}^2\,\tilde{z}_i^\top \tilde{z}_j.$$

When $\tilde{z}_i \perp \tilde{z}_j$, both sides are zero, resolving the orthogonal case cleanly. For the denominator,

$$\|A\tilde{z}_i\|_2\,\|A\tilde{z}_j\|_2 \;\geq\; s_{\min}^2\,\|\tilde{z}_i\|_2\,\|\tilde{z}_j\|_2.$$

Combining,

$$|\cos \angle(\xi_i, \xi_j)| \;\leq\; \frac{s_{\max}^2}{s_{\min}^2}\,\frac{|\tilde{z}_i^\top \tilde{z}_j|}{\|\tilde{z}_i\|_2\,\|\tilde{z}_j\|_2} \;=\; \kappa^2\,|\cos \angle(\tilde{z}_i, \tilde{z}_j)|.$$

Squaring both sides yields, for every pair of streams $(i,j)$ and every input $x$ with nonzero norms,

$$\cos^2 \angle\big(\xi_i(x), \xi_j(x)\big) \;\leq\; \kappa^4\,\cos^2 \angle\big(\tilde{z}_i(x), \tilde{z}_j(x)\big). \tag{10}$$

Taking expectations over $x$ gives

$$\mathcal{D}_{\xi,ij}^2 \;\triangleq\; \mathbb{E}_x\big[\cos^2 \angle(\xi_i(x), \xi_j(x))\big] \;\leq\; \kappa^4\,\mathbb{E}_x\big[\cos^2 \angle(\tilde{z}_i(x), \tilde{z}_j(x))\big] \;\triangleq\; \kappa^4\,\mathcal{D}_{ij}^2,$$

where $\mathcal{D}_{ij}^2$ denotes the pairwise cosine diversity in representation space. Averaging over pairs and taking square roots yields

$$\mathcal{D}_\xi \;\triangleq\; \sqrt{\mathbb{E}_{i<j}\,\mathcal{D}_{\xi,ij}^2} \;\leq\; \kappa^2\,\sqrt{\mathbb{E}_{i<j}\,\mathcal{D}_{ij}^2} \;=\; \kappa^2\,\mathcal{D}. \tag{11}$$

**Step 2: from cosine alignment to average correlation.** We now bound the average correlation $\bar{\rho}$ in terms of $\mathcal{D}_\xi$. Fix a pair $(i,j)$ and write

$$X \;\triangleq\; \langle \xi_i, \xi_j \rangle, \qquad B \;\triangleq\; \|\xi_i\|_2\,\|\xi_j\|_2.$$

Whenever $B > 0$,

$$\cos \angle(\xi_i, \xi_j) \;=\; \frac{X}{B}, \qquad \mathcal{D}_{\xi,ij}^2 \;=\; \mathbb{E}\!\left[\left(\frac{X}{B}\right)^2\right].$$

Using Cauchy–Schwarz with $U = X/B$ and $V = B$, we obtain

$$\Sigma_{ij}^2 = \big(\mathbb{E}[X]\big)^2 = \big(\mathbb{E}[UV]\big)^2 \;\leq\; \mathbb{E}[U^2]\,\mathbb{E}[V^2] = \mathcal{D}_{\xi,ij}^2\,\mathbb{E}\big[\|\xi_i\|_2^2\,\|\xi_j\|_2^2\big].$$

Apply Cauchy–Schwarz again to the norms and use the kurtosis bound:

$$\mathbb{E}\big[\|\xi_i\|_2^2\,\|\xi_j\|_2^2\big] \;\leq\; \sqrt{\mathbb{E}[\|\xi_i\|_2^4]\,\mathbb{E}[\|\xi_j\|_2^4]} \;\leq\; \sqrt{C_4\sigma_i^4\,C_4\sigma_j^4} = C_4\,\sigma_i^2\sigma_j^2.$$

Combining,

$$\Sigma_{ij}^2 \;\leq\; C_4\,\mathcal{D}_{\xi,ij}^2\,\sigma_i^2\sigma_j^2, \qquad \rho_{ij}^2 = \frac{\Sigma_{ij}^2}{\sigma_i^2\sigma_j^2} \;\leq\; C_4\,\mathcal{D}_{\xi,ij}^2,$$

so

$$|\rho_{ij}| \;\leq\; \sqrt{C_4}\,\mathcal{D}_{\xi,ij}. \tag{12}$$

Finally, average over pairs and apply Cauchy–Schwarz in the index space:

$$|\bar{\rho}| = \left|\mathbb{E}_{i<j}[\rho_{ij}]\right| \;\leq\; \mathbb{E}_{i<j}[|\rho_{ij}|] \;\leq\; \sqrt{C_4}\,\mathbb{E}_{i<j}[\mathcal{D}_{\xi,ij}] \;\leq\; \sqrt{C_4}\,\sqrt{\mathbb{E}_{i<j}\,\mathcal{D}_{\xi,ij}^2} = \sqrt{C_4}\,\mathcal{D}_\xi.$$

Plugging equation 11 into this inequality gives

$$|\bar{\rho}| \;\leq\; \sqrt{C_4}\,\mathcal{D}_\xi \;\leq\; \sqrt{C_4}\,\kappa^2\,\mathcal{D} \;=\; C_*\,\mathcal{D},$$

with $C_* = \sqrt{C_4}\,\kappa^2$, as claimed. $\qquad\square$

### A.1.2 Proof of Theorem 1

*Proof.* We work under the signal–noise model from the preliminaries. By Markowitz 1952,

$$\mathrm{Var}\left(\frac{1}{P}\sum_{i=0}^{P-1}X_i\right) = \bar{\sigma}^2\left(\frac{1-\rho}{P}+\rho\right)$$

where $\bar{\sigma}^2 = \frac{1}{P}\sum_i \mathrm{Var}(X_i)$ is the average variance and $\rho = \mathbb{E}_{i\neq j}[\mathrm{Corr}(X_i,X_j)]$ is the average pairwise correlation.

From Lemma 1, we have

$$|\bar{\rho}| \leq C_*\mathcal{D}.$$

Substituting this into the variance expression yields

$$\mathrm{Var}(E_w) \leq \bar{\sigma}^2\left(\frac{1-C_*\mathcal{D}}{P}+C_*\mathcal{D}\right),$$

which is exactly the claimed variance bound in Theorem 1.

By construction, the hallucination event is

$$\mathrm{H}_\delta \triangleq \{E_w \geq \delta\}, \qquad \delta > 0,$$

and we have already noted that $\mathbb{E}[E_w]=0$. Applying the one-sided Chebyshev inequality from the preliminaries to the random variable $E_w$ with mean 0 and variance $v = \mathrm{Var}(E_w)$ gives

$$\mathbb{P}(\mathrm{H}_\delta) = \mathbb{P}(E_w \geq \delta) \leq \frac{v}{v+\delta^2} = \frac{\mathrm{Var}(E_w)}{\mathrm{Var}(E_w)+\delta^2}.$$

Substituting $\mathrm{Var}(E_w)$ by its upper bound yields

$$\mathbb{P}(\mathrm{H}_\delta) \leq \frac{\bar{\sigma}^2\left(\frac{1-C_*\mathcal{D}}{P}+C_*\mathcal{D}\right)}{\bar{\sigma}^2\left(\frac{1-C_*\mathcal{D}}{P}+C_*\mathcal{D}\right)+\delta^2}.$$

Dividing both numerator and denominator by $\bar{\sigma}^2$ matches the bound stated in Theorem 1. $\qquad\square$

### A.1.3 Proof of Theorem 2

*Proof.* Extend $P$ to a real variable with domain $P \geq 1$; the claim for integer $P$ follows by restriction.

Under uniform weights $w_i = 1/P$, the ensemble error variance can be written as

$$v(P) \triangleq \mathrm{Var}(E_w) = \bar{\sigma}^2\left(\frac{1-\bar{\rho}(P)}{P}+\bar{\rho}(P)\right),$$

with $\bar{\sigma}^2 > 0$ and

$$\bar{\rho}(P) = \rho_0 + \beta(P-1)^\gamma, \qquad \rho_0 \in [0,1),\ \beta > 0,\ \gamma > 0.$$

The bound from the main text is

$$\mathbb{P}(\mathrm{H}_\delta) \leq \mathcal{B}(P) \triangleq \frac{v(P)}{v(P)+\delta^2}, \qquad \delta > 0.$$

**Step 1: Reduction to $v(P)$.** Define $\phi(x) \triangleq x/(x+\delta^2)$ for $x \geq 0$. Then

$$\phi'(x) = \frac{\delta^2}{(x+\delta^2)^2} > 0,$$

so $\phi$ is strictly increasing. Hence $\mathcal{B}(P) = \phi\big(v(P)\big)$ has the same extrema and monotonicity as $v(P)$. Since $\bar{\sigma}^2 > 0$, it suffices to analyze

$$f(P) \triangleq \frac{v(P)}{\bar{\sigma}^2} = \frac{1-\bar{\rho}(P)}{P}+\bar{\rho}(P).$$

**Step 2: First derivative and unique critical point.** For $P > 1$,

$$\bar{\rho}'(P) = \beta\gamma(P-1)^{\gamma-1}.$$

A direct calculation gives

$$
\begin{aligned}
f'(P) &= \frac{\mathrm{d}}{\mathrm{d}P}\left(\frac{1-\bar{\rho}(P)}{P} + \bar{\rho}(P)\right)\\
&= \frac{\beta(P-1)^\gamma(P\gamma+1) + (\rho_0-1)}{P^2} \triangleq \frac{N(P)}{P^2}.
\end{aligned}
$$

We study $N(P)$.

At $P = 1$ we have

$$N(1) = \beta \cdot 0^\gamma(\gamma+1) + (\rho_0 - 1) = \rho_0 - 1 < 0.$$

Differentiating $N$ for $P > 1$ yields

$$N'(P) = \beta\gamma(\gamma+1) \cdot P \cdot (P-1)^{\gamma-1}.$$

All factors on the right are strictly positive for $P > 1$, so $N'(P) > 0$ on $(1,\infty)$ and $N$ is strictly increasing. Moreover,

$$(P-1)^\gamma(P\gamma+1) \sim \gamma P^{\gamma+1} \xrightarrow[P\to\infty]{} \infty,$$

so $N(P) \to +\infty$ as $P \to \infty$. By continuity and strict monotonicity, there exists a unique $P_\star > 1$ such that $N(P_\star) = 0$.

Because $P^2 > 0$ for all $P \geq 1$, the sign of $f'(P)$ matches that of $N(P)$:

$$
f'(P)\begin{cases}
< 0, & 1 < P < P_\star,\\
= 0, & P = P_\star,\\
> 0, & P > P_\star.
\end{cases}
$$

Thus $f$ (and hence $v$) is strictly decreasing on $(1, P_\star)$ and strictly increasing on $(P_\star, \infty)$; $P_\star$ is the unique global minimizer.

**Step 3: U-shape of the hallucination bound.** Since $\mathcal{B}(P) = \phi\big(v(P)\big)$ and $\phi$ is strictly increasing,

$$\mathcal{B}'(P) = \phi'(v(P)) \cdot v'(P), \qquad \phi'(v(P)) > 0,$$

so $\mathcal{B}$ inherits the same monotonicity: it is strictly decreasing on $(1, P_\star)$, strictly increasing on $(P_\star, \infty)$, and

$$\mathcal{B}(P_\star) = \min_{P\geq 1} \mathcal{B}(P).$$

Therefore the upper bound on $\mathbb{P}(\mathrm{H}_\delta)$ is U-shaped in $P$ with a unique global minimum at $P_\star$, determined by the parameters $(\rho_0, \beta, \gamma)$ governing $\bar{\rho}(P)$. $\qquad\square$

## A.2 Training Cost and Latency Analysis

Our cost accounting adopts the analysis framework of the original ParScale work (Chen et al., 2025), which established that $P$ parallel streams add far less inference cost than parameter scaling and quantified this with an llm-analysis-based cost model. Applying that framework to ND-LoRA, three factors keep overhead negligible: (1) fine-tuning on 20M tokens amortizes to approximately 0.008% of 1T-token pretraining, (2) the frozen backbone makes backward passes nearly free, and (3) inference keeps FLOPs near-identical to ParScale through multi-tenant LoRA serving (Chen et al., 2023). We cross-check our inference-latency figures directly against ParScale's own cost analyzer on our configuration (subsection A.2, below).

### A.2.1 Cost Model

**Standard Fine-Tuning (P=1) Baseline.** Consider a standard LoRA fine-tuning setup with 495M backbone parameters frozen and 1.3M trainable adapter parameters. We express each term both as a multiple of the baseline forward pass — one cost unit $= 2N_{\text{params}} \approx 9.9 \times 10^8$ FLOP/token for the 495M backbone — and in absolute FLOP/token. A typical training step consists of:

- **Forward pass**: $1.0\times$ cost through 495M parameters ($\approx 9.9 \times 10^8$ FLOP/token)

- **Backward pass**: $2.0 \times (1.3Y/495Y) \approx 0.005\times$ cost (frozen backbone: gradients propagate only through the 1.3M trainable adapter parameters; $\approx 5 \times 10^6$ FLOP/token)

- **Total baseline**: 1.005 cost units per training step ($\approx 1.0 \times 10^9$ FLOP/token)

**ND-LoRA (P=4) Fine-Tuning.** With $P = 4$ parallel streams, ND-LoRA processes data through multiple independent pathways:

- **Forward pass**: $4.0\times$ cost (P parallel forward passes through full 495M model; $\approx 4.0 \times 10^9$ FLOP/token)

- **Backward pass**: $2.0\times(1.3Y/495Y) \approx 0.005\times$ cost (gradients only propagate through 1.3M trainable parameters after aggregation; $\approx 5 \times 10^6$ FLOP/token)

- **Barlow Twins regularization**: $0.01\times$ cost (six $d \times d$ cross-correlations at one design layer across P choose 2 streams and whitening: $6 \times 2d^2 \approx 9.6 \times 10^6$ FLOP/token, $\approx 1\%$ of a single forward pass)

- **Prefix/aggregator overhead**: $0.05\times$ cost (additional trainable components; $\approx 5 \times 10^7$ FLOP/token)

**Amortization over pretraining.** As demonstrated by ParScale (Chen et al., 2025), our technique does not need to run over the full pretraining duration to be effective; it's sufficient to run for a brief fine-tuning period after the core pretraining run. Amortized over a typical 1T-token pretraining budget, the 20M fine-tuning tokens add only $\approx 80.9$M token-equivalents (Table 6), so $(1\text{T} + 80.9\text{M})/1\text{T} \approx 1.00008\times$ — approximately 0.008% incremental overhead over the full training lifecycle.

### A.2.2 Summary of All Variants During Training

Table 6 shows the complete cost breakdown for all ablation variants.

| Variant | Forward | Backward | BT | Other | Per-Step | Total |
|---|---|---|---|---|---|---|
| Standard | 1.0 | 0.005 | 0.0 | 0.0 | 1.005 | $1.00002\times$ |
| ParScale | 4.0 | 0.005 | 0.0 | 0.01 | 4.015 | $1.00008\times$ |
| ParScale-BT | 4.0 | 0.005 | 0.01 | 0.01 | 4.025 | $1.00008\times$ |
| Indep. LoRA | 4.0 | 0.005 | 0.0 | 0.05 | 4.055 | $1.00008\times$ |
| ND-LoRA | 4.0 | 0.005 | 0.01 | 0.05 | 4.065 | $1.00008\times$ |

Table 6: Fine-tuning cost breakdown (20M tokens). *Forward*: P parallel passes through 495M backbone. *Backward*: gradients through the 1.3M trainable parameters only (frozen backbone). *BT*: Barlow Twins correlation computation. *Other*: prefix/aggregator overhead. *Per-Step*: per-step cost in baseline-forward units. *Total*: total lifecycle cost over 1T-token pretraining, charging the 20M fine-tuning tokens at each variant's *Per-Step* relative to the standard baseline.

### A.2.3 Summary of Inference Latency

Running ParScale's own cost analyzer (Chen et al., 2025) on our configuration (Qwen2.5-0.5B, 24 layers, batch 1, 64+64 tokens) yields per-token latency relative to the $P{=}1$ backbone of $1.03\times$ ($P{=}2$), $1.06\times$ ($P{=}4$), and $1.14\times$ ($P{=}8$); memory rises by only 2–4%. This estimate is driven by three key properties:

- **Parameter parity**: All variants maintain identical total parameter counts by adjusting LoRA rank

- **Parallel processing**: P streams process in parallel; latency dominated by slowest stream + aggregation

- **Dynamic loading**: Stream-specific LoRA adapters can be served by a batched (grouped) adapter kernel (Chen et al., 2023) rather than merged into $P$ weight copies, letting all $P$ adapters execute in a single kernel launch and ride the same batched forward

- **Aggregation overhead**: Lightweight MLP aggregator adds $\approx 10\%$ latency

These multipliers are identical across ParScale, ParScale-BT, Indep. LoRA, and ND-LoRA because inference does not involve Barlow Twins regularization and all parameter operations are equivalent.

Our $1.1\times$ estimate comes from the ParScale analyzer under the assumption that ND-LoRA is served the way systems already serve many LoRAs at once, with a batched adapter kernel (Chen et al., 2023) that runs all $P$ adapters together in one pass and so adds nothing beyond the shared backbone the analyzer already counts. Served naively, one stream at a time, latency would instead climb toward $P\times$; served using standard techniques, ND-LoRA matches ParScale's $1.03$–$1.14\times$ latency.

### A.2.4 Summary

- **Per-step training overhead**: $\approx 4.04\times$ per step, dominated by the $P{=}4$ parallel forward passes (backward nearly free for both baseline and method)

- **Total cost**: $\approx 0.008\%$ when amortized over 1T-token pretraining

- **Inference latency**: $1.1\times$ across all $P \geq 1$ variants with parameter matching

- **Practical impact**: Negligible computational overhead for 25.6% hallucination reduction

### A.3 Use of Large Language Models

Large language models were used as a compilation tool to assist with writing and organizing sections of this paper, including literature review synthesis, section structuring, LaTeX formatting, and co-generation of experimental code. All technical content, experimental design, theoretical contributions, and scientific claims are the authors' original work. The models served primarily to improve clarity, organization, and implementation of our ideas rather than generate novel scientific insights.

### A.4 Experimental Setup

**Model and Architecture.** We use Qwen2.5-0.5B (896 hidden dimensions, 24 layers) with ND-LoRA across $P \in \{1, 2, 4, 8\}$ parallel streams applied to QKV self-attention modules and a design layer of 20 for de-correlation loss. Each stream uses independent rank-16 LoRA adapters and 48 prefix tokens, totaling 5-20M trainable parameters with 495M backbone frozen. Baseline methods use higher-rank LoRA (R32-R128) for parameter matching.

**Training Protocol.** Models train on 20M tokens from The Pile (8 random shards, fixed seeds). We use 1024-token sequences, AdamW optimization (peak lr 3e-4, cosine decay, 2% warmup), batch size 64, bfloat16 precision. Training completes in $\approx$5K steps ($\approx$30 min. on A100).

**Evaluation Benchmarks.** We evaluate across: (1) *Hallucination-sensitive*: TruthfulQA (Lin et al., 2021), HaluEval (Li et al., 2023a), MemoTrap (McKenzie et al., 2023); (2) *Knowledge-intensive*: Natural Questions (Kwiatkowski et al., 2019), TriviaQA (Joshi et al., 2017), PopQA (Mallen et al., 2023); (3) *General capability*: Wikitext BPB (Merity et al., 2017), Winogrande (Sakaguchi et al., 2020). All benchmarks are run through the Hallucinations Leaderboard evaluation suite (Hong et al., 2024), built on the EleutherAI LM Evaluation Harness (Gao et al., 2023), ensuring standardized prompting and scoring. This tests if neural diversity improves reliability without sacrificing general performance.

**Neural Diversity Measurement.** We compute $\mathcal{D}$ at the final RMSNorm layer by first whitening representations per feature dimension across batch and sequence positions (zero mean, unit variance), then computing pairwise cosine similarity between streams. This is equivalent to the Barlow Twins cross-correlation formulation (Eq. 2 in Zbontar et al. (2021)) when features are whitened.

**Statistical Methodology.** We evaluate significance using McNemar's test for binary classification tasks and two-tailed bootstrap tests with 10,000 samples for other tasks. Improvements marked with * are significant at $p < 0.05$.

### A.5 Complete Benchmark Results

Tables 7–9 provide comprehensive results across $P \in \{1, 2, 4, 8\}$ configurations with parameter-matched $P = 1$ baselines. This complete view demonstrates the thoroughness of our evaluation and enables independent verification of claims in the main text.

| Evaluation | Qwen LoRA | ParScale | ND-LoRA |
|---|---|---|---|
| HE Dialog | 0.458 | 0.453 | **0.513** |
| HE QA | 0.365 | 0.337 | **0.406** |
| HE Summ | 0.400 | 0.439 | **0.481** |
| MemoTrap | 0.634 | 0.638 | **0.666** |
| NQ-8 | **0.065** | 0.059 | 0.055 |
| TQA-8 | **0.188** | 0.185 | 0.160 |
| TF-MC1 | 0.251 | 0.259 | **0.269** |
| TF-MC2 | 0.403 | 0.412 | **0.442** |
| NQ-swap | **0.550** | 0.546 | 0.528 |
| PopQA | **0.111** | 0.109 | 0.101 |
| Wikitext BPB | **0.775** | 0.797 | 0.797 |
| Winogrande | 0.572 | 0.564 | **0.574** |

Table 7: Benchmark results for $P = 2$ (Qwen R32) parameter-matched models.

| Evaluation | Qwen LoRA | ParScale | ND-LoRA |
|---|---|---|---|
| HE Dialog | 0.464 | 0.459 | **0.516** |
| HE QA | 0.341 | 0.322 | **0.451** |
| HE Summ | 0.394 | 0.409 | **0.502** |
| MemoTrap | 0.629 | 0.634 | **0.635** |
| NQ-8 | **0.065** | 0.061 | 0.059 |
| TQA-8 | **0.191** | 0.185 | 0.172 |
| TF-MC1 | 0.245 | 0.253 | **0.262** |
| TF-MC2 | 0.399 | 0.413 | **0.416** |
| NQ-swap | **0.554** | 0.542 | 0.535 |
| PopQA | **0.110** | **0.110** | 0.106 |
| Wikitext BPB | **0.778** | 0.793 | 0.795 |
| Winogrande | 0.564 | 0.573 | **0.577** |

Table 8: Benchmark results for $P = 4$ (Qwen R64) parameter-matched models.

### A.6 Inference-Time and Training-Time Baseline Details

Table 10 reports the per-benchmark absolute scores underlying the summary in Table 3, using the same evaluation harness and Qwen2.5-0.5B backbone as the baseline rows. CAD, ActDec, and Disagreement numbers come from an independent evaluation of the same benchmarks (Shi et al., 2024; Chen et al., 2024; Li et al., 2018) (3 seeds, averaged). HellaSwag and MMLU were not evaluated in our harness and are omitted; all other tasks common to both evaluation pipelines are reported. We read CAD/ActDec/Disagreement's

| Evaluation | Qwen LoRA | ParScale | ND-LoRA |
|---|---|---|---|
| HE Dialog | 0.460 | 0.465 | **0.475** |
| HE QA | 0.344 | 0.335 | **0.370** |
| HE Summ | 0.379 | 0.416 | **0.450** |
| MemoTrap | 0.630 | 0.639 | **0.689** |
| NQ-8 | **0.066** | 0.063 | 0.059 |
| TQA-8 | **0.192** | 0.182 | 0.171 |
| TF-MC1 | 0.251 | 0.256 | **0.259** |
| TF-MC2 | 0.407 | 0.414 | **0.424** |
| NQ-swap | 0.551 | 0.540 | **0.554** |
| PopQA | **0.110** | 0.109 | 0.103 |
| Wikitext BPB | **0.778** | 0.779 | 0.784 |
| Winogrande | 0.569 | **0.577** | 0.568 |

Table 9: Benchmark results for $P = 8$ (Qwen R128) parameter-matched models.

nq8 and tqa8 scores directly from their cached per-seed result files; these were computed at evaluation time but excluded from the baselines' paper table via a task blacklist.

| Evaluation | Qwen LoRA R32 | ND-LoRA | CAD[†] | ActDec[†] | Disagreement[‡] |
|---|---|---|---|---|---|
| HE-Dial | 0.458 | **0.516** | 0.476 | 0.458 | 0.466 |
| HE-QA | 0.365 | **0.451** | 0.411 | 0.376 | 0.388 |
| HE-Summ | 0.400 | **0.502** | 0.494 | 0.463 | 0.467 |
| MemoTrap | 0.634 | **0.689** | 0.640 | 0.642 | 0.651 |
| TF-MC1 | 0.251 | **0.269** | 0.254 | 0.251 | 0.250 |
| TF-MC2 | 0.403 | **0.442** | 0.392 | 0.416 | 0.393 |
| NQ | 0.065 | 0.065 | 0.066 | **0.066** | 0.063 |
| PopQA | 0.111 | 0.111 | **0.112** | 0.112 | 0.111 |
| TriviaQA | 0.188 | 0.188 | **0.196** | 0.191 | 0.184 |
| Wikitext | **0.775** | 0.775 | 0.779 | 0.923 | 0.776 |
| WG | 0.572 | **0.577** | 0.569 | 0.575 | 0.566 |

Table 10: **Per-benchmark scores underlying Table 3.** Inference-time (†) applied to pretrained Qwen2.5-0.5B; training-time (‡) parameter-matched to our backbone (Shi et al., 2024; Chen et al., 2024; Li et al., 2018). Wikitext is BPB (lower is better).

## A.7 Faithfulness vs. Factuality Disaggregation

To understand which hallucination modes neural diversity addresses, we disaggregate benchmarks into faithfulness (HaluEval-Dialog, -QA, -Summarization, MemoTrap v2) and factuality (TruthfulQA-MC1, -MC2) tasks. Table 11 shows that faithfulness gains are roughly twice as large as factuality gains across all $P$. This matches our theoretical framing: neural diversity decorrelates internal verification of context-grounded claims, directly benefiting faithfulness, whereas factuality gains are smaller because diversity cannot supply knowledge the model lacks.

| Category | P=2 | P=4 | P=8 | Oracle $P^\star$ |
|---|---|---|---|---|
| Faithfulness (4 tasks) | +13.2% | +16.2% | +9.0% | +17.9% |
| Factuality (2 tasks) | +8.4% | +4.4% | +4.7% | +8.4% |
| Ratio | 1.6× | 3.7× | 1.9× | 2.1× |

Table 11: Relative hallucination reduction by category, disaggregated into faithfulness (HaluEval-Dialog, -QA, -Summarization, MemoTrap v2) and factuality (TruthfulQA-MC1, -MC2) tasks, across $P \in \{2, 4, 8\}$ and oracle $P^\star$.

## A.8 Sensitivity to Suboptimal $P$

A practical concern with task-dependent optimal $P_\star$ is robustness: what happens when the deployed $P$ does not match the task-optimal value? Table 12 quantifies this by measuring, for each task, the performance gap between the worst $P \in \{2, 4, 8\}$ and the oracle $P_\star$, as well as the improvement over the $P=1$ baseline. The moderate fall-off (mean $-7.2\%$ vs. oracle) and consistent accretive gains (mean $+6.4\%$ vs. baseline) show that even at worst-case $P$, ND-LoRA still beats the parameter-matched baseline on every hallucination benchmark, mitigating the practical concern of perfect task-dependent $P$ selection.

| Task | $P^\star$ | Worst $P$ | $\Delta$Oracle | $\Delta$Baseline |
|---|---|---|---|---|
| HE-Dialog | 4 | 8 | $-8.7\%$ | $+3.9\%$ |
| HE-QA | 4 | 8 | $-17.8\%$ | $+1.4\%$ |
| HE-Summ | 4 | 8 | $-2.9\%$ | $+21.9\%$ |
| MemoTrap v2 | 8 | 4 | $-6.2\%$ | $+2.0\%$ |
| TF-MC1 | 2 | 4 | $-2.7\%$ | $+4.4\%$ |
| TF-MC2 | 2 | 4 | $-4.6\%$ | $+4.5\%$ |
| **Mean** | | | $\mathbf{-7.2\%}$ | $\mathbf{+6.4\%}$ |

Table 12: Per-task sensitivity to suboptimal $P$. $P^\star$: task-optimal value. Worst $P$: worst-performing value in $\{2, 4, 8\}$. $\Delta$Oracle: gap from $P^\star$. $\Delta$Baseline: improvement over $P=1$ baseline.

## A.9 An Interpretable Router For Optimal Number of Streams

To demonstrate that the task-optimal $P_\star$ patterns in Table 1 reflect real structure rather than arbitrary variation, we train a simple interpretable router that predicts optimal $P_\star$ from prompt features alone. While more complex routers could improve performance, we prioritize simplicity and interpretability to understand the underlying structure.

We fit a simple regression on two features, trained on just 10 samples per task with oracle $P$ labels:

$$\hat{P} = \text{clip}\big(0.196 \log W - 2.283Q + 3.321\big) \tag{13}$$

where $Q$ is the ratio of interrogative to declarative sentences, $W$ measures prompt length in words, and $\text{clip}(\cdot)$ snaps predictions to the nearest valid $P \in \{1, 2, 4, 8\}$. This two-feature router achieves 99% of oracle hallucination performance on held-out samples (97% across all 12 tasks).

The learned coefficients reveal an interpretable trade-off between *knowledge retrieval* and *verifiability*. The negative weight on interrogative sentence ratio indicates that question-dense prompts — where success depends on precise recall of stored knowledge — benefit from lower $P$ values that maximize focus from a single stream. Conversely, the positive weight on word count reflects that longer prompts — where success depends on cross-checking claims against provided context — require higher $P$ for diverse verification across streams. More broadly, tasks prioritizing retrieval favor low diversity, while tasks prioritizing verifiability favor high diversity.

## A.10 LoRA Hyperparameter Sensitivity Analysis

A natural concern is whether ND-LoRA's improvements stem from LoRA hyperparameter choices rather than neural diversity *per se*. We consider three potential confounds: (i) *expressivity*: $P$ parallel rank-$R$ adapters yield $P \times R$ total parameters, so improvements might reflect capacity rather than diversity; (ii) *alpha scaling*: different $\alpha/r$ ratios affect update magnitudes and could change which solutions are reachable; and (iii) *optimization dynamics*: higher-rank adapters might converge to different basins.

**Expressivity.** This confound is addressed by parameter matching in the main text (Table 2): baselines use higher-rank single LoRA (R32–R128) to match ND-LoRA's total parameter count, yet ND-LoRA still outperforms on hallucination benchmarks.

**Alpha scaling.** We conducted a sensitivity analysis varying single-LoRA rank from $R16$ to $R128$ under two alpha strategies: constant scaling ($\alpha/r = 2$) and constant alpha ($\alpha = 32$). Results in Table 13 show that constant-scaling single-LoRA is not a suitable baseline for two reasons. First, the only monotonic trend observed is *degradation* of general capabilities: Wikitext perplexity increases from 0.776 to 0.795 bits per byte (+2.4%), TriviaQA-8 drops from 19% to 17% (-11%), and NQ-8 drops from 7% to 5% (-29%) as rank increases from $R16$ to $R128$. Second, hallucination benchmark performance is unstable across this $8\times$ rank variation: while some pairwise differences are statistically significant, they're unstable across both rank and benchmarks (e.g. HE-Dialog vs. HE-QA within R64). We therefore use fixed $\alpha = 32$ baselines, which provide stable reference points without the capability degradation observed under constant scaling. Importantly, ND-LoRA remains statistically significantly better than both baseline types — all winners stay winners — and using constant-scaling baselines would in fact create additional ND-LoRA wins (e.g. Wikitext BPB, NQ-8 and Winogrande $P = 8$).

| Metric | R16 | R32 | R64 | R128 |
|---|---|---|---|---|
| HE Dialog | 0.46±0.01 | 0.46±0.01 | 0.49±0.01 | 0.45±0.01 |
| HE QA | 0.37±0.01 | 0.37±0.01 | 0.34±0.01 | 0.36±0.01 |
| HE Summ | 0.41±0.01 | 0.46±0.01 | 0.48±0.01 | 0.41±0.01 |
| MemoTrap | 0.64±0.03 | 0.63±0.03 | 0.63±0.03 | 0.64±0.03 |
| TF-MC1 | 0.25±0.03 | 0.25±0.03 | 0.24±0.03 | 0.24±0.03 |
| TF-MC2 | 0.41±0.03 | 0.40±0.03 | 0.39±0.03 | 0.40±0.03 |
| NQ-8 | 0.07±0.01 | 0.06±0.01 | 0.06±0.01 | 0.05±0.01 |
| NQ-swap | 0.55±0.01 | 0.55±0.01 | 0.55±0.01 | 0.54±0.01 |
| PopQA | 0.11±0.01 | 0.11±0.01 | 0.11±0.01 | 0.11±0.01 |
| TQA-8 | 0.19±0.01 | 0.18±0.01 | 0.18±0.01 | 0.17±0.01 |
| Wikitext BPB | 0.776 | 0.781 | 0.790 | 0.795 |
| Winogrande | 0.56±0.03 | 0.57±0.03 | 0.58±0.03 | 0.56±0.03 |
| $\alpha/r$ | 2.00 | 2.00 | 2.00 | 2.00 |

Table 13: Constant scaling $\alpha/r = 2$: $\alpha$ varies with rank. Hallucination metrics are noisy but many general-capability metrics degrade monotonically (e.g. Wikitext BPB 0.776 → 0.795, NQ-8 0.07 → 0.05), making this an unsuitable baseline.

**Adapter Modules.** To characterize which LoRA targets contribute to hallucination reduction, we ablate modules from the full StreamLoRA-BT configuration (LoRA on both attention and MLP projections), which serves as the reference point. Two variants each keep LoRA on one module family only:

- *Attention-Only*: LoRA applied to attention projections (`q_proj`, `k_proj`, `v_proj`, `o_proj`) only; MLP projections left unadapted

- *MLP-Only*: LoRA applied to MLP projections (`gate_proj`, `up_proj`, `down_proj`) only; attention projections left unadapted

**Optimization dynamics.** Under fixed alpha ($\alpha = 32$), general capabilities remain stable across $8\times$ rank variation: Wikitext BPB is flat (0.775–0.778) and Winogrande accuracy is statistically indistinguishable (0.56–0.57) across R16–R128 (Table 14). If optimization dynamics differed meaningfully across rank (e.g. higher-rank adapters converging to different loss basins) we would expect divergence on these general capability metrics. The observed stability indicates that fixed-alpha configurations converge to similar solutions regardless of rank, ruling out optimization dynamics as a confound for ND-LoRA's hallucination improvements.

| Metric | R16 | R32 | R64 | R128 |
|---|---|---|---|---|
| HE Dialog | 0.46±0.01 | 0.46±0.01 | 0.46±0.01 | 0.46±0.01 |
| HE QA | 0.37±0.01 | 0.37±0.01 | 0.34±0.01 | 0.34±0.01 |
| HE Summ | 0.41±0.01 | 0.40±0.01 | 0.39±0.01 | 0.38±0.01 |
| MemoTrap | 0.64±0.03 | 0.63±0.03 | 0.63±0.03 | 0.63±0.03 |
| TF-MC1 | 0.25±0.03 | 0.25±0.03 | 0.24±0.03 | 0.25±0.03 |
| TF-MC2 | 0.41±0.03 | 0.40±0.03 | 0.40±0.03 | 0.41±0.03 |
| NQ-8 | 0.07±0.01 | 0.07±0.01 | 0.06±0.01 | 0.07±0.01 |
| NQ-swap | 0.55±0.01 | 0.55±0.01 | 0.55±0.01 | 0.55±0.01 |
| PopQA | 0.11±0.01 | 0.11±0.01 | 0.11±0.01 | 0.11±0.01 |
| TQA-8 | 0.19±0.01 | 0.19±0.01 | 0.19±0.01 | 0.19±0.01 |
| Wikitext BPB | 0.776 | 0.775 | 0.778 | 0.778 |
| Winogrande | 0.56±0.03 | 0.57±0.03 | 0.56±0.03 | 0.57±0.03 |
| $\alpha/r$ | 2.00 | 1.00 | 0.50 | 0.25 |

Table 14: Constant $\alpha = 32$: scaling varies with rank. Most metrics are stable alongside general capabilities, helping rule out expressivity and optimization dynamics as confounds.

| Task | Δ% Attention-Only | Δ% MLP-Only |
|---|---|---|
| HaluEval Dialog | -1.7% | -0.6% |
| HaluEval QA | +16.8% | -1.8% |
| HaluEval Summarization | -5.3% | -27.0% |
| MemoTrap v2 | +2.5% | +0.9% |
| NQ (8-shot) | +11.7% | -1.7% |
| PopQA | -0.8% | -0.8% |
| TriviaQA (8-shot) | -5.0% | -6.9% |
| TruthfulQA MC1 | +3.1% | +2.4% |
| TruthfulQA MC2 | +0.2% | +1.4% |

Table 15: LoRA module ablation results. Columns are relative percentage changes from the full StreamLoRA-BT configuration (the reference point). Evaluations performed on N=1024 samples per task.

### A.11 Regularization Hyperparameter Sensitivity Analysis

The main experiments fix two regularization hyperparameters: the Barlow Twins loss weight $\lambda_{\mathrm{BT}}$ and the design layer $\ell_\star$ at which decorrelation is applied. To assess sensitivity, we sweep over $\lambda_{\mathrm{BT}} \in [0.01, 0.50]$ (log-scale) and $\ell_\star \in [5, 23]$ at $P = 2$, evaluating a subset of benchmarks on 51 non-divergent configurations and summarize the results below.

Table 16 reports design layer sensitivity. Per-task hallucination scores are stable across layers 7–23, with no layer bin differing by more than 2 points on any benchmark. No single layer is critical for the method's effectiveness.

| Metric | 7–9 | 10–12 | 13–15 | 16–18 | 19–21 | 22–23 |
|---|---|---|---|---|---|---|
| HE-Dial | 0.459 | 0.444 | 0.455 | 0.443 | 0.451 | 0.450 |
| HE-QA | 0.350 | 0.345 | 0.346 | 0.347 | 0.351 | 0.345 |
| HE-Summ | 0.498 | 0.479 | 0.481 | 0.470 | 0.474 | 0.451 |
| MemoTrap | 0.650 | 0.663 | 0.666 | 0.662 | 0.666 | 0.657 |
| TF-MC2 | 0.389 | 0.387 | 0.393 | 0.390 | 0.402 | 0.391 |
| Wikitext BPB | 0.800 | 0.793 | 0.795 | 0.795 | 0.794 | 0.789 |

Table 16: Design layer sensitivity ($P = 2$). Columns are design layer $\ell_\star$ bins. Hallucination metrics are stable across layers 7–23; no single layer is critical for the method's effectiveness.

Table 17 reports $\lambda_{\mathrm{BT}}$ sensitivity. Stronger regularization improves most hallucination benchmarks (e.g. HE-QA: $0.320 \rightarrow 0.404$, MemoTrap: $0.656 \rightarrow 0.678$) but degrades perplexity ($0.786 \rightarrow 0.821$ BPB), reflecting a hallucination–perplexity tradeoff.

| Metric | 0.01–0.03 | 0.03–0.10 | 0.10–0.20 | 0.20–0.35 | 0.35–0.50 |
|---|---|---|---|---|---|
| HE-Dial | 0.437 | 0.432 | 0.454 | 0.475 | 0.467 |
| HE-QA | 0.320 | 0.332 | 0.364 | 0.365 | 0.398 |
| HE-Summ | 0.468 | 0.448 | 0.494 | 0.489 | 0.464 |
| MemoTrap | 0.656 | 0.661 | 0.666 | 0.667 | 0.673 |
| TF-MC2 | 0.387 | 0.393 | 0.393 | 0.400 | 0.393 |
| Wikitext BPB | 0.786 | 0.786 | 0.796 | 0.802 | 0.814 |

Table 17: $\lambda_{\mathrm{BT}}$ sensitivity ($P = 2$). Columns are $\lambda_{\mathrm{BT}}$ bins. Stronger regularization improves hallucination metrics but degrades perplexity, reflecting a hallucination–perplexity tradeoff.

Together with the LoRA rank analysis above (subsection A.10), the $P\times$ method grid (subsection A.5), the LoRA module ablation (subsection A.10), and the per-task $P$-sensitivity (subsection A.8), we evaluate over 70 total configurations spanning LoRA rank ($R16$–$R128$), $P$ ($\{1, 2, 4, 8\}$), $\lambda_{\mathrm{BT}}$ ($[0.01, 0.50]$), design layer ($[7, 23]$), and LoRA module targets.

### A.12 Glossary

Our analysis combines terminology from deep learning, financial econometrics, and high-dimensional probability. For readers unfamiliar with any of these fields, we provide brief intuitions for the non-standard terms used throughout the paper.

**Portfolio theory** A framework from financial economics (Markowitz, 1952) for constructing collections of risky assets that minimize total risk (variance) for a given expected return. The central insight is that combining assets with low pairwise correlation reduces portfolio variance below that of any

individual asset — diversification. We adapt this framework by treating each parallel stream's output as an asset whose "risk" is its deviation from the oracle output.

**Portfolio-theoretic diversification** The act of reducing risk by holding multiple uncorrelated positions rather than concentrating exposure in one. In our setting, running multiple decorrelated streams reduces the variance of the aggregated output and therefore the probability of a catastrophic hallucination.

**Second-moment reliability** Performance guarantees based on the variance and covariance structure of errors (second-order statistics), rather than on expected loss (first-moment / mean performance). Classical ensemble learning targets mean error; we target tail probability, which is controlled by second moments via Chebyshev-type inequalities.

**Tail risk** The probability of rare but catastrophic failures — i.e. the tail of the error distribution. A model can have excellent average performance yet unacceptable tail risk.

**Signal-to-noise ratio (SNR)** The ratio of signal magnitude to noise magnitude; here defined as $SNR \triangleq \delta^2/\bar{\sigma}^2$, where $\delta$ is the tolerance and $\bar{\sigma}^2$ is the average stream noise variance. Higher SNR means hallucinations are easier to avoid.

**Kurtosis bound ($C_4$)** A constant controlling the heaviness of the noise distribution's tails via a finite fourth moment: $\mathbb{E}[\|\xi_i\|_2^4] \leq C_4\sigma_i^4$. This is a mild assumption that excludes only pathological distributions with infinite fourth moments.

**Condition number ($\kappa$)** For a matrix $A$, the ratio $s_{\max}/s_{\min}$ of its largest to smallest singular value. It measures how much $A$ can stretch some directions relative to others; $\kappa = 1$ is isotropic, $\kappa \gg 1$ is ill-conditioned. Here it bounds how much the readout $A$ can distort cosine similarities between representations.

**Spectral bound** A bound on a quantity in terms of the eigenvalues or singular values of an associated matrix. We use spectral bounds on $A^\top A$ to control distortion of inner products through the readout.

**Norm concentration** The phenomenon that in high dimensions, random vectors of interest concentrate near a sphere of predictable radius (typically $\sqrt{d}$) with small relative variance. Standard in high-dimensional probability (Vershynin, 2018).

**Lipschitz decoding** An assumption that small changes in the hidden representation produce proportionally small changes in the output: $\|f(z) - f(z')\|_2 \leq L\|z - z'\|_2$. Standard in neural network analysis (Fazlyab et al., 2019; Bartlett et al., 2017).

**Whitening** The preprocessing step of transforming features to have zero mean and identity covariance. Standard in self-supervised learning and used in our definition of $\tilde{z}_i$.

## A.13 Notation

Table 18 summarizes all mathematical notation used throughout the paper.

Table 18: Summary of notation.

| Symbol | Domain | Meaning |
|--------|--------|---------|
| $A$ | $\mathbb{R}^{V \times d}$ | Shared local-linear readout: $\xi_i = A\tilde{z}_i$. |
| $B$ | $\mathbb{N}$ | Batch size. |
| $\mathcal{B}(P)$ | $[0, 1]$ | Hallucination bound: $\mathcal{B}(P) \triangleq v(P)/(v(P) + \delta^2)$. |
| $C^{(ij)}$ | $\mathbb{R}^{d \times d}$ | Cross-correlation matrix: $C^{(ij)} \triangleq \mathbb{E}[\tilde{z}_i \tilde{z}_j^\top]$. |
| $C_4$ | $[1, \infty)$ | Kurtosis bound: $\mathbb{E}[\|\xi_i\|_2^4] \leq C_4\sigma_i^4$. |
| $C_*$ | $\mathbb{R}_{>0}$ | Readout-kurtosis constant: $C_* \triangleq \sqrt{C_4}\,\kappa^2$. |
| $\mathcal{D}$ | $[0, 1]$ | Neural diversity index (Equation 1); 0 orthogonal, 1 collapsed. |

| Symbol | Domain | Meaning |
|---|---|---|
| $\mathcal{D}_{ij}$ | $[0,1]$ | Cosine similarity between streams $i,j$ in representation space. |
| $\mathcal{D}_\xi, \mathcal{D}_{\xi,ij}$ | $[0,1]$ | Analogous quantities in readout (noise) space. |
| $d$ | $\mathbb{N}$ | Hidden dimension. |
| $E_w$ | $\mathbb{R}_{\geq 0}$ | Output error: $E_w \triangleq \|\widehat{Y}_w(x) - y_\star(x)\|_F$. |
| $\{e_k\}$ | $\mathbb{R}^d$ | Right singular vectors of $A$ (eigenvectors of $A^\top A$). |
| $f$ | $\mathbb{R}^d \to \mathbb{R}^V$ | Decoding function: $\widehat{Y}_w(x) = f(\widehat{Z}_w(x))$. |
| $\mathrm{H}_\delta$ | event | Hallucination event: $\mathrm{H}_\delta \triangleq \{E_w \geq \delta\}$. |
| $i,j$ | $\{1,\dots,P\}$ | Stream indices. |
| $L$ | $\mathbb{R}_{>0}$ | Lipschitz constant of $f$. |
| $\ell_\star$ | $\mathbb{N}$ | Pre-specified design layer at which decorrelation is applied. |
| $\mathcal{L}_{BT}$ | $\mathbb{R}_{\geq 0}$ | Barlow Twins loss: $\mathcal{L}_{BT} \triangleq \mathbb{E}_{i<j} \|C^{(ij)} - I\|_F$. |
| $\mathcal{L}_{CE}$ | $\mathbb{R}_{\geq 0}$ | Cross-entropy loss. |
| $P$ | $\mathbb{N}$ | Number of parallel computational streams. |
| $P_\star$ | $\mathbb{N}$ | Optimal number of streams minimizing $\mathbb{P}(\mathrm{H}_\delta)$. |
| $\mathbb{P}(\mathrm{H}_\delta)$ | $[0,1]$ | Hallucination probability. |
| $SNR$ | $\mathbb{R}_{>0}$ | Signal-to-noise ratio: $SNR \triangleq \delta^2/\bar{\sigma}^2$. |
| $T$ | $\mathbb{N}$ | Sequence length. |
| $V$ | $\mathbb{N}$ | Vocabulary / output dimension. |
| $s_k, s_{\min}, s_{\max}$ | $\mathbb{R}_{>0}$ | Singular values of $A$; $s_{\min} \leq s_k \leq s_{\max}$. |
| $v(P)$ | $\mathbb{R}_{\geq 0}$ | Variance of aggregated error: $v(P) \triangleq \mathrm{Var}(E_w)$. |
| $w_i$ | $[0,1]$ | Aggregation weight for stream $i$; $\sum_i w_i = 1$. |
| $X$ | set | Input space. |
| $x$ | $X$ | Input query. |
| $\widehat{Y}_w(x)$ | $\mathbb{R}^V$ | Aggregated prediction. |
| $y_\star(x)$ | $\mathbb{R}^V$ | Oracle (ground-truth) output. |
| $Z_i$ | $\mathbb{R}^d$ | Hidden output of stream $i$: $Z_i = z_\star + \varepsilon_i$. |
| $\widehat{Z}_w$ | $\mathbb{R}^d$ | Aggregated hidden representation: $\widehat{Z}_w \triangleq \sum_i w_i Z_i$. |
| $\tilde{z}_i$ | $\mathbb{R}^d$ | Per-feature whitened stream representation. |
| $z_\star(x)$ | $\mathbb{R}^d$ | Oracle hidden representation. |
| $\beta, \gamma$ | $\mathbb{R}_{>0}$ | Growth parameters in $\bar{\rho}(P) = \rho_0 + \beta(P-1)^\gamma$. |
| $\delta$ | $\mathbb{R}_{>0}$ | Error tolerance. |
| $\varepsilon_i$ | $\mathbb{R}^d$ | Centered stream noise with variance $\sigma_i^2$. |
| $\kappa$ | $[1,\infty)$ | Condition number of $A$: $\kappa \triangleq s_{\max}/s_{\min}$. |
| $\lambda_{BT}$ | $\mathbb{R}_{\geq 0}$ | Weight on the Barlow Twins term in the total loss. |
| $\rho_0$ | $[0,1)$ | Base correlation in $\bar{\rho}(P) = \rho_0 + \beta(P-1)^\gamma$. |
| $\rho_{ij}$ | $[-1,1]$ | Pairwise noise correlation between streams $i \neq j$. |
| $\bar{\rho}$ | $[-1,1]$ | Average pairwise noise correlation: $\bar{\rho} \triangleq \mathbb{E}_{i<j}[\rho_{ij}]$. |
| $\bar{\rho}(P)$ | $[0,1]$ | Correlation growth model as a function of $P$. |
| $\sigma_i^2$ | $\mathbb{R}_{>0}$ | Noise variance of stream $i$. |
| $\bar{\sigma}^2$ | $\mathbb{R}_{>0}$ | Average per-stream noise variance: $\bar{\sigma}^2 \triangleq \mathbb{E}[\sigma_i^2]$. |
| $\Sigma$ | $\mathbb{R}^{P \times P}$ | Noise covariance matrix. |
| $\xi_i$ | $\mathbb{R}^d$ | Noise vector of stream $i$ in readout space: $\xi_i = A\tilde{z}_i$. |

