# OpenReview forum: "Neural Diversity Regularizes Hallucinations in Language Models"
_TMLR — Decision pending for TMLR_

### Review · Reviewer_LNHo · 2026-04-08

**Summary Of Contributions:**

The authors introduce the idea of 'neural diversity' — decorrelated parallel representations — as a principled mechanism that can be empirically demonstrated to reduce the occurrence rate of certain categories of hallucinations. They also provide the first formal tail bounds for hallucination probability in ensembled language models.

**Audience:**

Yes

**Audience Explanation:**

* This is an important research area; people care a lot about hallucination detection, and hallucinations are one of the core challenges of deploying LLMs in safety-critical settings.
* The specifics of the implementation would be of interest to researchers in this area; however, there is no code release as far as I can see, so I cannot count this in the work's favor without an explicit commitment by the authors to release the code.
* The work's algorithmic contributions are neither trivial nor obvious; building on classical deep ensembles, which have been shown to exploit disagreement/diversity across predictors for robustness and uncertainty, the paper integrates more recent contributions such as ParScale, LoRA and Barlow-Twins-style decorrelation in an innovative fashion, making this aspect of the work a sound and useful contribution.
* The ablation result showing that the decorrelation term helps beyond the parallel-stream backbone, and that it combines super-linearly with stream-specific LoRA, is a useful contribution.

**Broader Impact Concerns:**

None.

**Claims And Evidence:**

No

**Claims Explanation:**

* The abstract somewhat overstates the scope of the empirical claims. As written, "reduces hallucination rates" sounds like a general statement about language models, but the evidence in the paper is much narrower: the reported gains are for a specific architecture/training recipe (ND-LoRA with parallel streams and decorrelation regularization), on a selected set of hallucination-oriented benchmarks, under fixed parameter and data budgets. I would strongly suggest qualifying this claim throughout, e.g. "reduces hallucination rates on the evaluated benchmarks/in our setting" rather than implying a general reduction result. It would also be helpful to have a clearer story about when, and why, this particular method helps.
* The paper derives tail bounds for an ensemble-like / parallel-stream construction under a stylized signal-noise model and several nontrivial assumptions. The proof strategy relies on averaging correlated streams and importing a portfolio/diversification argument, so it is not obvious how much of the result transfers to ordinary single-stream LMs. This would be a helpful addition.
* The abstract and theory use “hallucination” as if it were a single construct. However, the benchmark suite operationalizes at least two different and potentially contradictory ones, faithfulness to context and factuality of response. Faithfulness and factuality can come apart whenever the provided context is wrong, incomplete, outdated, or adversarial. It is not clear under which definition of hallucination this method purports to improve model capability.
* The paper frames task-dependent optimal diversity as a positive scientific finding, from a practical standpoint it also appears to introduce an additional task-specific hyperparameter. The paper would benefit from clarifying the fall-off effects of suboptimal choices here and it should be explicitly referenced as a limitation.

**Requested Changes:**

Please address the weaknesses outlined in the above comments.

---

> ### Author Response · Authors · 2026-04-09
>
> We thank the reviewer for a careful, constructive, and technically informed reading of our work. Every point raised is well-taken, and we believe the resulting revisions substantially strengthen the paper. We address each point below.
>
> **(1) Overclaimed abstract and clearer motivation.** We have better scoped claims throughout, e.g. ensembled language models, on evaluated benchmarks, in our experimental settings. We also clarify when and why neural diversity helps: the subset of hallucinations from correlated representational errors — not missing knowledge — are noise amenable to diversification (Section 1).
>
> **(2) Transfer to single-stream LMs.** Our current work focuses on language models in parallelized (self-ensembled) settings. Preliminary work suggests portfolio-theoretic dynamics emerge in single-stream LMs via internal parallel structures (e.g. multi-head attention), but co-exist with interference dynamics requiring a distinct theoretical treatment (anonymous, in preparation). We reference this as future work and outline the open questions.
>
> **(3) Faithfulness vs. factuality.** We especially appreciate this distinction — it is both conceptually and empirically important, especially as we observe inverse scaling in some faithfulness measures like MemoTrap [1]. We’ve added a disaggregated analysis (new Appendix table) separating faithfulness tasks (HaluEval-Dialog, -QA, -Summarization, MemoTrap v2) from factuality tasks (TruthfulQA-MC1, -MC2):
>
> | Category | P=2 | P=4 | P=8 | Oracle P* |
> |---|---|---|---|---|
> | Faithfulness (4 tasks) | +13.2% | +16.2% | +9.0% | +17.9% |
> | Factuality (2 tasks) | +8.4% | +4.4% | +4.7% | +8.4% |
> | Ratio | 1.6× | 3.7× | 1.9× | 2.1× |
>
> Although factuality improves materially, faithfulness gains are consistently $\approx 2\times$ larger, which is theoretically expected: neural diversity decorrelates internal verification of context-grounded claims, directly benefiting faithfulness. In contrast, factuality may require knowledge the model lacks, which our approach explicitly does not attempt to address and thus limits diversity-based gains. We clarify this in Sections 4.2 and 7.
>
> [1] McKenzie et al., "Inverse Scaling: When Bigger Isn't Better," Transactions on Machine Learning Research, 2023.
>
> **(4) Task-dependent optimal diversity as a hyperparameter.** We agree this adds complexity and now state this as an explicit consideration in the Discussion. However, three observations mitigate the practical concern.
>
> *(4a) Fall-off is moderate but always accretive.* A new sensitivity table (Appendix) shows that even at worst-case $P$, ND-LoRA still beats the parameter-matched baseline on every benchmark:
>
> | Task        | P*  | Worst P | ΔOracle   | ΔBaseline |
> |-------------|-----|---------|-----------|-----------|
> | HE-Dialog   | 4   | 8       | −8.7%     | +3.9%     |
> | HE-QA       | 4   | 8       | −17.8%    | +1.4%     |
> | HE-Summ     | 4   | 8       | −2.9%     | +21.9%    |
> | MemoTrap v2 | 8   | 4       | −6.2%     | +2.0%     |
> | TF-MC1      | 2   | 4       | −2.7%     | +4.4%     |
> | TF-MC2      | 2   | 4       | −4.6%     | +4.5%     |
> | **Mean**    |     |         | **−7.2%** | **+6.4%** |
>
> *(4b) Default works well.* $P{=}4$ captures 97% of oracle hallucination performance with no tuning (Section 5.4).
>
> *(4c) Simple router achieves near-optimality.* The two-feature router in Section A.7 achieves 99% of oracle hallucination performance: $\hat{P} = \text{clip}[0.196 \log W - 2.283Q + 3.321]$ for prompt length in words $W$ and interrogative to declarative sentence ratio $Q$.
>
> The coefficients connect directly to point (3): the negative weight on interrogative ratio routes factuality-dependent prompts to lower $P$ (where neural diversity helps less), while the positive weight on prompt length routes context-grounded prompts to higher $P$ (where faithfulness gains are largest). This convergence between the router's learned structure and the hallucination decomposition suggests the task-dependent optimum reflects a retrieval-vs-verifiability tradeoff rather than arbitrary sensitivity.
>
> **(5) Code release.** We will release both model checkpoints, training code and evaluation code to reproduce all results in our paper in the camera-ready version.
>
> We are grateful for the reviewer's engagement. We believe these revisions — scoping claims, disaggregating faithfulness and factuality, analyzing fall-off sensitivity, and connecting the router to the hallucination decomposition — substantially strengthen the paper's clarity and scientific contribution.

---

> > ### Comment · Reviewer_LNHo · 2026-04-22
> >
> > My thanks to the authors for a careful and detailed rebuttal. The rebuttal essentially addresses all of the concerns I had, so I am content to accept the paper with these revisions.

---

> > > ### Author Response · Authors · 2026-04-22
> > >
> > > Dear Reviewer LNHo,
> > >
> > > Thank you for the thoughtful engagement throughout and for supporting acceptance. The faithfulness-vs-factuality distinction you drew out in particular sharpened the paper and we're grateful for it.
> > >
> > > Best,
> > > The authors

---

### Review · Reviewer_F4D3 · 2026-04-10

**Summary Of Contributions:**

This paper theoretically characterizes the LLM hallucination using a second-moment reliability problem. Based on this framework, the theoretical result gives the hallucination probability and provides the non-nmonotonicity in reliability scaling. Moreover, the theroetical prediction aligns well with empirical experiment. The empirical experiment on Qwen2.5 model is convincing and supports the theoretical results well.

**Audience:**

Yes

**Audience Explanation:**

Yes, this result is very interesting and I believe individuals in TMLR's audience will be interested in knowing the findings of this paper.

**Claims And Evidence:**

Yes

**Claims Explanation:**

This work presents two main theoretical results. One is the Theorem 1, it provides the upper bound of the tail hallucination probability with respect to the neural diversity index. The result reveals the realtion between the the important metric neural diversity index proposed in this paper and the halluciation probability. This result is validated by connecting the upper bound of $\bar{\rho}$ and the  tail hallucination probability in Figure 3 and reveals statistically significant connection.

Another is the Theorem 2, it presents the U-shaped behavior of the hallucination bound with respect to P. This result is validated in Figure 1 with a U-shaped curve where performance peaks at optimal P* then degrades. (However, Figure 1 doesn't look like a U-shaped).

**Requested Changes:**

It seems not to be very clear in Figure 1. In its caption, it indicates that "we find a U-shaped curve where performance peaks at optimal P.". The curve looks like a $\cap$ not U. It should be stated more clear here.

Also, this paper doesn't provide sufficient explaination when it introduces some concept for the first time. E.g. " a second-moment reliability problem" , " portfolio-theoretic bound" , "parallel representations" , etc. It makes the paper hard to read.

Notations are not clearly defined. E.g. Theorem 1. C-star, D, and P are not defined. A theorem should be a complete statement instead of simply an upper bound.

Also, on Page 15, "By the spectral bounds on $A^\top A ... ". This upper bound is incorrect if x is orthogonal to y. I have no idea what is x and y and I don't know if they will really satisfy this condition.

---

> ### Author Response · Authors · 2026-04-16
>
> Dear Reviewer F4D3,
>
> Thank you for the careful reading and supportive assessment. We appreciate the specific, actionable feedback and address each point below (and in an accompanying minor revision).
>
> **Figure 1: "U-shaped" language.** You're right, the figure shows an inverted-U for *reliability*, while the U-shape applies to *hallucination probability*. We've clarified the caption so that reliability is described as following an inverted-U (peaking at optimal $P_\star$, then degrading), with our theorems predicting the corresponding U-shape in hallucination probability (Fig. 1).
>
> **First-use concept explanations.** Given that we borrow terminology from other fields (e.g. financial econometrics) that might be unfamiliar to some readers, we've added brief remarks in the main body (e.g. an introductory sentence in section 1 framing hallucinations as "a form of noise amenable to portfolio-theoretic diversification") and also added a new Glossary appendix (sign-posted from the top of section 2) for fuller on intuition "second-moment reliability," "portfolio-theoretic," and other cross-field terminology.
>
> **Notation clarity.** We did a full notation pass across the paper and added a comprehensive notation table (Appendix A.13) listing every symbol used in the paper for global reference. For Theorem 1 specifically, we've added inline definitions of $\mathcal{D}$ (neural diversity index, Eq. 1), $P$ (number of parallel streams), $C_\ast = \sqrt{C_4}\,\kappa^2$ (constant depending on kurtosis bound and readout condition number), and $SNR = \delta^2/\bar{\sigma}^2$ directly in the theorem statement.
>
> **Spectral bound when $x \perp y$.** Thank you — this was a proof presentation issue. The step $|x^\top A^\top A\, y| \leq \|A^\top A\|_{\mathrm{op}} |x^\top y|$ holds in our specific setting under a mild condition noted in the motivating prose in Section 2.2 ("has one underlying model with aligned neuron-level representations"), which we now formally declare, spell out intermediate proof steps and clarify that $x, y$ are different stream representations.
>
> Specifically, the preliminaries now include feature alignment as assumption (iii): parallel streams project onto each readout feature with matching sign. Without it, mixed per-feature signs let cancellation shrink the unweighted pre-readout inner product while the readout's asymmetric stretching keeps the weighted sum large. With it, per-feature contributions add constructively, no cancellation occurs, and the bound holds uniformly. This holds in our setting in two ways:
> 1. for pre-regularization parallel architectures (e.g. ParScale), streams share a backbone and combine via convex aggregation, preserving feature sign across streams — you can empirically verify this with $\mathcal{D} \sim 1$ (near-identity) in Table 5 for standard parallel architectures; and,
> 2. for regularized models (e.g. ND-LoRA), Barlow Twins regularization reinforces $C^{(ij)} \to I$, driving same-feature diagonals to one while suppressing cross-feature leakage.
>
> The clarification is worked out in the Appendix proof of Lemma 1; the conclusions of Lemma 1 are unchanged.
>
> These revisions — the clarified Fig. 1 framing, feature alignment assumption, and the glossary and notation audits — have all been incorporated and meaningfully improved the paper's precision. Thank you again for the careful reading that surfaced them.

---

> > ### Comment · Reviewer_F4D3 · 2026-04-18
> >
> > Thanks for the revision! The updated notations and tables are much more helpful for understanding this  paper and the theoretical statements look much much formal.  I also checked the clarified proof (e.g. x and y are associated with the eigenvectors) and it seems to be correct from my perspective. I am satisfied with the current version. Given that all claims are correctly supported, I would like to support the acceptance of this work.

---

> > > ### Author Response · Authors · 2026-04-22
> > >
> > > Dear Reviewer F4D3,
> > >
> > > Thank you for the careful re-reading and for supporting acceptance. We appreciated the specificity of your initial feedback — it led to real improvements in the paper's precision.
> > >
> > > Best,
> > > The authors

---

### Review · Reviewer_ZWGc · 2026-04-13

**Summary Of Contributions:**

**Summary:**
The paper proposes reducing LLM hallucinations by encouraging neural diversity through decorrelated parallel representations. Drawing on portfolio theory, the authors derive tail bounds on hallucination probability and introduce ND-LoRA, a fine-tuning method that combines parallel LoRA adapters with Barlow Twins regularization. Experiments across several benchmarks report an average improvement of  about $14\$%. However, the empirical evidence is limited, making it difficult to draw strong or generalizable conclusions.


**Strengths:**
1. I really like the authors’ reframing hallucination as a second-moment (variance/correlation) problem rather than a first-moment (mean accuracy) problem. It is conceptually elegant and underexplored till date. The connection to Markowitz portfolio theory is creative, and is a genuinely valuable contribution.
2. The paper correctly highlights a key limitation of existing approaches (e.g., RAG, RLHF, CD): improving average performance does not control tail risks, meaning models can still produce severe errors despite strong mean accuracy.
3. The theoretical framework is well-structured and aligns reasonably with the experimental observations.
4. The evaluation spans multiple datasets and benchmarks.

**Weakness:**
1. The scale of experiments is very very limited. All results are based on a Qwen-500M model trained on only 20M tokens from The Pile. This has several limitations:
    1. Without any results on large models (atleast 7B, to 13B), the paper’s claim that neural diversity is a *‘third axis of scaling’* is hard to accept. The correlation structures, collapse dynamics, and diversity indices may behave very differently at scale.
    2. Representation geometry and correlation structure can differ significantly at scale; so conclusions from small models may not transfer. Moreover, observed hallucination behavior may also be confounded by small-model limitations such as the softmax bottleneck.
2. There is substantial lack of hyperparameter exploration. Neither the training hyperparamters nor LoRA rank is ablated thoroughly, leaving open the possibility that results are sensitive to these choices.
3. Lemma 1 assumes a shared linear readout ($\xi_i = A\tilde{z}_i$), but ND-LoRA uses stream-specific adapters, implying different readouts. It is unclear whether the bound $|\bar{\rho}| \leq C*D$ still holds when A varies across streams, or whether the condition number $\kappa$ remains meaningful in this setting.
4. The paper does not compare against inference-time scaling methods (e.g., self-consistency, temperature scaling, speculative decoding), despite discussing them.

**Audience:**

Yes

**Audience Explanation:**

The findings are useful for researchers studying hallucinations and small langauge models.

**Claims And Evidence:**

Yes

**Claims Explanation:**

Check the strengths section.

**Requested Changes:**

I urge the authors to take a look at the weakness provided in section 1 and revise their manuscript accordingly.

---

> ### Author Response · Authors · 2026-04-16
>
> Dear Reviewer ZWGc,
>
> Thank you for a thoughtful and technically incisive review. We especially appreciate the recognition of the second-moment reframing via portfolio theory and the observation that mean-accuracy methods leave tail risk uncontrolled. Weaknesses (2)–(4) are incorporated in the revision (including new head-to-head comparisons); (1) motivates scoping changes.
>
> **(1) Scale and "third axis of scaling."** We take this point seriously. Rather than overreach to a single 7B run that would remain underpowered for a scaling claim, we have tightened the paper's scope to match exactly the evidence we present. (As an independent lab, we are financially constrained from running the multi-size, multi-seed sweep a rigorous scaling claim would require, which reinforces our preference to scope claims to what the evidence supports rather than speculate beyond it.) We no longer invoke scaling-axis language and describe the empirical study as a small-scale demonstration of the broader framework, with extrapolation to 7B+ flagged as future work.
>
> **(2) Inference-time baselines.** We've added a new subsection *Comparison with Other Methods* benchmarking ND-LoRA against CAD [Shi et al. 2024], ActDec [Chen et al. 2024], and matched training-time baseline Disagreement Regularization [Li et al. 2018]:
>
> | Method | Type | Halluc. Δ% | Knowledge Δ% |
> |---|---|---|---|
> | **ND-LoRA** | integrated | **+14.6%** | +0.2% |
> | CAD | inference-time | +4.1% | +1.2% |
> | ActDec | inference-time | +1.5% | −2.6% |
> | Disagreement | training-time | +1.7% | −1.1% |
>
> In short, ND-LoRA delivers $\approx 3.5\times$ the hallucination gain of the next-best baseline while holding knowledge within 0.2% of the $P{=}1$ baseline, and wins every per-benchmark hallucination column in the appendix. Self-consistency and temperature scaling diversify over sampling noise from a fixed representation, while ND-LoRA diversifies the representations themselves; the two attack different stages of the pipeline, and we flag ND-LoRA $+$ self-consistency as multiplicative future work in the Discussion.
>
> Speculative decoding matches the target distribution and makes no reliability claim; we removed it from the implied baselines.
>
> **(3) Lemma 1 shared readout.** We've added a remark to clarify this modeling simplification. The shared-readout simplification is architecturally anchored: post-design computation factors as a frozen backbone composed with a single `lm_head`, both shared across streams ($A \in \mathbb{R}^{V \times d}$). Only the rank-$16$ LoRA adapters in post-design layers perturb the readout — pre-design adapters shape $\tilde{z}_i$, not $A$ — giving $A_i = A + \Delta A_i$ with $\mathrm{rank}(\Delta A_i) \le 16 \ll d = 896$, a rank-$16$ correction to a rank-$d$ readout. Section 3.1 documents this as a theory–architecture bridge alongside the other simplifications (linearization, whitening, norm concentration). The $R^2 = 0.943$ fit across $P \in \{1, 2, 4, 8\}$ with operationally stream-specific adapters is consistent with these contributions remaining small in practice.
>
> **(4) Hyperparameter sensitivity.** We've added a new subsection *Hyperparameter Sensitivity* in Mechanistic Analysis, summarizing three sweeps that were already in the Appendix (but not visible in the main text) and adding three new ones:
>
> | Axis | Range | Finding |
> |---|---|---|
> | LoRA rank + $\alpha$ | $R16$–$R128$; $\alpha/r{=}2$, $\alpha{=}32$ | Neither rank nor $\alpha$ scaling is a confound |
> | Design layer | $\ell_\star \in [7, 23]$ | Gain stable across layer choice |
> | $\lambda_\text{BT}$ | $[0.01, 0.50]$ log | Hallucination–perplexity tradeoff |
> | $P$ | $\{2, 4, 8\}$ | Worst-choice $P$ still beats the parameter-matched baseline |
> | LoRA modules | Attention vs. MLP | Attention-only outperforms MLP-only LoRA |
>
> We are grateful for your engagement. The direct inference-time baseline comparison, stream-specific extension, scoped claims, and surfaced sensitivity summary all directly respond to your concerns, and we believe the paper is materially stronger for them.

---

### Decision · Action_Editor_5vDv · 2026-06-15

**Recommendation:** Accept with minor revision

**Additional Comments:**

Primarily, I believe the authors can improve the stated claims in the paper to be specific to one single small language model, i.e. Qwen 2.5. Specifically, I request the following minor revisions:
- Be explicit about the experimental evidence reported and claims made in the paper.
    - In the abstract — stating “in our setting” is not enough. Be direct and explicit about what the setting is.
    - In the contributions listed towards the end of the introduction (second bullet point), explicitly state that a single SLM was used for evaluation and highlight which one.
    - Be clear at the beginning of Section 4 (before Sec 4.1) that only one SLM was tested.
- The authors claim that their proposed approach enables reliability gains without scaling compute. Can the authors also comment on the time/cost required to run their approach (towards the end of the abstract and end of the introduction)? What are the tradeoffs? Consider adding this in the discussion.
- As promised, the authors must release model checkpoints, training code and evaluation code to reproduce all results in the submission.

**Audience:**

Yes

**Audience Explanation:**

The problem statement is of interest and is an important research area --- hallucination detection is a core challenge for deploying language models in safety-critical settings (including deployment of SLMs in resource-constrained settings). All reviewers agree that the relevant findings of this paper will be of interest to the TMLR community. I support this assessment.

**Claims And Evidence:**

Yes

**Claims Explanation:**

The paper proposes reducing language model hallucinations by encouraging neural diversity through decorrelated parallel representations. Drawing on portfolio theory, the authors derive tail bounds on hallucination probability in ensembled language models and introduce ND-LoRA, a fine-tuning method that combines parallel LoRA adapters with Barlow Twins regularization. Their principled approach is empirically demonstrated to reduce the occurrence rate of a subset of hallucinations derived from correlated representational errors (and operates primarily on instruction following / faithfulness to context behavior) for the Qwen 2.5-0.5B model.

The authors’ reframing of hallucination as a second-moment (variance/correlation) problem rather than a first-moment (mean accuracy) problem is conceptually elegant, well-structured, highlights a key limitation of existing approaches, and aligns reasonably with experimental observations [ZWGc]. The algorithmic contributions and results of the paper are creative [ZWGc], very interesting [F4D3], sound, and a useful contribution [LNHo].

All reviewers' requested changes have been addressed in the rebuttal/discussion phase and I concur with their support for acceptance of this work to TMLR conditional on minor writing revisions.

---

> ### Author Response · Authors · 2026-07-10
>
> Thank you, we're grateful for your careful handling of the paper and for the reviewers' generous engagement throughout. Your framing of the revisions was clear and easy to act on. The camera-ready incorporates all of your requested changes:
>
> * **Scope made explicit.** The abstract now names the setting directly (Qwen2.5-0.5B, 20M Pile tokens, 12 tasks); the second contribution bullet and the opening of Section 4 both state that the empirical results come from a single small language model, while the theory remains model-agnostic.
> * **Cost and tradeoffs surfaced.** We report the compute and latency cost (+0.008% pretraining, 1.1× inference latency) at the close of the abstract and introduction, and name latency as the sole recurring tradeoff in the discussion. The appendix now gives a sharper, FLOP-level accounting: it borrows ParScale's cost-analysis framework, cross-checks our inference latency directly against their own analyzer (1.03–1.14× for P=2–8), and clarifies that near-identical inference FLOPs hold under multi-tenant LoRA serving — with the modest adapter overhead stated explicitly.
> * **Artifacts released.** Training, evaluation & analysis code and model checkpoints to reproduce all key results have been made public on GitHub and linked in the paper.
>
> Thank you again for the thoughtful review process — it's been a pleasure working with you and the reviewers.

---

> > ### Comment · Action_Editor_5vDv · 2026-07-20
> >
> > Thanks to the authors for addressing the requested changes. I request a couple of more minor changes from the authors to approve the camera ready version:
> > - Please cite the Pile dataset that you mention in the abstract and appendix A.4.
> > - While 8 evaluation benchmarks are listed in Appendix A.4 and 12 total are mentioned in the abstract, introduction and are listed in Tables 7 & 8, please be explicit about how there are 12 tasks total and why are they important. Appendix A.4 might be a good place to make this clear with a reference to this part of the appendix in the main paper where necessary.